# Three-dimensional surface motion capture of multiple freely moving pigs using MAMMAL

Liang An [1], Jilong Ren[2,3], Tao Yu[1,4], Tang Hai [2,3] ✉, Yichang Jia [5,6,7] ✉ & Yebin Liu [1,8] ✉

Understandings of the three-dimensional social behaviors of freely moving large-size mammals are valuable for both agriculture and life science, yet challenging due to occlusions in close interactions. Although existing animal pose estimation methods captured keypoint trajectories, they ignored deformable surfaces which contained geometric information essential for social interaction prediction and for dealing with the occlusions. In this study, we develop a Multi-Animal Mesh Model Alignment (MAMMAL) system based on an articulated surface mesh model. Our self-designed MAMMAL algorithms automatically enable us to align multi-view images into our mesh model and to capture 3D surface motions of multiple animals, which display better performance upon severe occlusions compared to traditional triangulation and allow complex social analysis. By utilizing MAMMAL, we are able to quantitatively analyze the locomotion, postures, animal-scene interactions, social interactions, as well as detailed tail motions of pigs. Furthermore, experiments on mouse and Beagle dogs demonstrate the generalizability of MAMMAL across different environments and mammal species.

Pigs, like other animals, express their general well-being through their various behaviors, such as locomotion, body postures, interactions with environments, and communications with peers[1]. For example, tail motions reflect their emotional states[2,3], and limb dynamics give us clues about their health conditions[4–6]. Quantitatively monitoring pigs' behaviors is important for both the welfare of pigs and agriculture, because health condition is indispensable for pork production[5]. Moreover, as an animal model, pigs have been proven to be important for life science research[6–11]. For some neurological disease modeling cases, pigs demonstrate better performance than previously widely-used rodent models[6], because of their closer genetics, brain anatomy, and physiology to humans. Therefore, behavioral recording is crucial for the understanding of

neurobiological processes[12]. For example, modeling movement-related brain disorders (e.g., Huntingtin Disease[6,7], Amyotrophic Lateral Sclerosis[13,14], Parkinson Disease[10], etc.) requires quantification of locomotion and 3D postural dynamics of pigs, while modeling cognition-related brain disorders (e.g., Alzheimer's Disease[15], Autism[16,17], depression[17], etc.) requires quantification of animal-scene interactions or animal-animal social behaviors. Although studies on pig behavior recognition have been achieved on monocular videos[2–5,18–30], a system for accurate three-dimensional (3D) markerless pig motion reconstruction and quantitative analysis is currently lacking.

Three-dimensional deformable surfaces provide essential geometric information for individuals to visually sense each other and

[1]Department of Automation, Tsinghua University, Beijing, China. [2]State Key Laboratory of Stem Cell and Reproductive Biology, Institute of Zoology, Chinese Academy of Sciences, Beijing, China. [3]Beijing Farm Animal Research Center, Institute of Zoology, Chinese Academy of Sciences, Beijing, China. [4]Tsinghua University Beijing National Research Center for Information Science and Technology (BNRist), Beijing, China. [5]School of Medicine, Tsinghua University, Beijing, China. [6]IDG/McGovern Institute for Brain Research at Tsinghua, Beijing, China. [7]Tsinghua Laboratory of Brain and Intelligence, Beijing, China. [8]Institute for Brain and Cognitive Sciences, Tsinghua University, Beijing, China. ✉e-mail: haitang@ioz.ac.cn; yichangjia@tsinghua.edu.cn; liuyebin@mail.tsinghua.edu.cn

further interact with each other in a social interaction context. However, the 3D surface information has not been captured by most existing laboratory animal motion capture methodologies, including SLEAP[31,32] and DeepLabCut[33,34], which estimate animal motions by sparse two-dimensional (2D) keypoints. These approaches gain accurate and efficient localization of visible keypoints but often confuse individual animals due to the severe motion feature invisibility caused by mutual occlusions. Currently, multi-view cameras have been applied to integrate 2D keypoints into 3D representation of a single animal through triangulation[35–39], which minimizes the 2D keypoint projection error. However, the 2D to 3D conversion is not trivial especially for multiple animals because of two obstacles: 1) how can multi-view unordered 2D cues be spatially and temporally associated? 2) how can 3D postures be robustly reconstructed from occluded 2D poses? The first obstacle has not been addressed by currently available animal motion capture methodologies. In addition, the second obstacle is not easy to overcome by direct triangulation because a keypoint is often invisible in closely interacting conditions. Recently, regression-based methods such as DANNCE[40] and BEV[41] were employed to handle self-occlusions to some degree[40,42]; however, they required large-scale 3D datasets for machine training, which is both time- and money-consuming. Therefore, a toolbox that can economically track multiple animal 3D geometric information is urgently required.

In this paper, we developed a Multi-Animal Mesh Model Alignment system, referred to as MAMMAL, to reconstruct the 3D surface motion of freely moving pigs in their natural living environments. MAMMAL includes three sequential stages: Detection, Detection Matching, and Mesh Fitting. Built upon an articulated mesh model of pig (the PIG model), MAMMAL overcame the above two obstacles with self-designed cross-view matching and mesh fitting algorithms. Consequently, MAMMAL endowed us with the ability to perform analysis on various previously concerned pig behaviors[1–5,18–27,30] in 3D space, including locomotion, postures, animal-scene interactions, and social behaviors. Besides, MAMMAL works well in different experimental settings, for example, in different numbers and sizes of pigs. Moreover, thorough evaluations affirmed the higher accuracy of MAMMAL over previous methods for animal 3D social pose estimation. For example, we quantitatively analyzed tail motions for pigs in different social hierarchies, which have been associated with their emotions[3]. We also showed that MAMMAL tracked mouse extremities with a competitive accuracy compared to DANNCE, and could be generalized to other large-size mammal species like Beagle dogs.

## Results

### Surface motion capture of multiple pigs using MAMMAL

Our MAMMAL system enables us to capture the motions of multiple pigs in their natural living environments (Fig. 1a, Supplementary Fig. 1). To achieve this goal, we established an articulated surface mesh model, called the PIG model, which contains 11239 vertices that are driven by 62 joints (Fig. 1b). These 62 joints can animate majority of pig motions we videotaped, including motions of body, tail, jaw, ears, and toes (Supplementary Fig. 2a, b, Supplementary Movie 1). They represent the ultimate motion freedom of the PIG model. Each joint is controlled by a 3 DOF (degrees of freedom) rotation vector, the changes of which would affect the geometric position of both joints and vertices of the PIG model. Among these joints, 24 crucial joints were used for trunk/leg motion control by ignoring motions on the tail, ears, or toes to efficiently assess pig locomotion and truck social contact without loss of accuracy (Fig. 1b, Supplementary Fig. 2c). In our PIG model, we include 19 easily-accessible keypoints, which are located on the nose, eyes, ears, tail root, center of mass, and legs, similar to that described in other 2D toolboxes[43,44] (Supplementary Fig. 3). To correspond our 2D observations to the PIG model, we defined 3D positions of these 19 keypoints in the PIG model (i.e., 3D keypoints).

These 3D keypoints were mapped from the joints or vertices of the mesh model, each of which could be the position of a joint or a vertex, or the interpolation of several joints/vertices. By aligning the 2D keypoints detected by our multiple cameras to the PIG model, MAMMAL directly optimizes the rotations of the joints together with pig scales and 6 DOF (the freedom of movement of a rigid body in 3D space) to place an individual in 3D space with arbitrary poses (Fig. 1c).

As the first stage of Detection, MAMMAL leveraged two deep neural networks PointRend[45] and HRNet[46] to produce the 2D cues. Different from DeepLabCut or SLEAP, MAMMAL first generated the bounding box and silhouette of each pig instance, then normalized each pig image region to a fixed 384 × 384 resolution before detecting visible keypoints (Supplementary Fig. 4a). The resolution normalization ensured equally good performance on different view angles, including side views from which the pigs occupied extremely larger areas than that from corner views. To train the deep neural networks, we manually annotated a BamaPig2D dataset, which contained 3340 images and 11504 pig instances with thoroughly-labeled bounding boxes, silhouettes, and keypoints of each individual (Supplementary Fig. 4b, c). After MAMMAL Detection, the unordered multi-view 2D cues were applied for surface alignment based on our PIG model (Supplementary Movie 2). The second stage was Detection Matching, which associated unordered temporal and spatial 2D cues with their corresponding objects (Supplementary Fig. 5a). At the initial timepoint ($T = 0$), we designed an innovative cross-view graph matching algorithm to effectively match spatially unordered 2D cues. At successive timepoints ($T > 0$), the PIG model enabled the identification of both visible and invisible keypoints to ensure the success of tracking. The third stage was Mesh Fitting, in which the PIG model provided important surface information to deal with severe occlusions (Supplementary Fig. 5b). To this end, the matched 2D cues were aligned to the PIG model to produce the 3D keypoints and surface geometric information of each individual, which allows the surface information to be simultaneously rendered into each view to determine the occlusion relationship (Supplementary Fig. 5b). The occlusion relationship and surface information guided the filtering of broken silhouettes and wrong keypoints when animals were overlapped to ensure good fitting quality (Supplementary Fig. 5c, d). Taking advantage of the surface representation power of the PIG model, MAMMAL estimates multiple animal mesh with invisible keypoints (Fig. 1d) and yields fine-grained 3D surface motion of multiple pigs in a social interaction context (Fig. 1e, Supplementary Fig. 6).

### MAMMAL enables various pig behavioral analyses

MAMMAL provides us the ability to analyze various pig behaviors. (i) Animal-Scene interaction measurement. MAMMAL enables 3D scene-aware behavior recognition, which automatically determines the drinking, feeding, and other desired states of a pig by using 3D motion capture results and scene priors (Fig. 2a–c, Supplementary Movie 3). (ii) Posture discovery, an important topic in brain research[32,33,38]. To address it, we curated 44 short motion clips from 4 individual pigs, which included 20819 poses, and clustered them using t-SNE (Fig. 2d). By manually checking the poses at the local density peaks, we identified 8 distinct postures (Fig. 2e, Supplementary Movie 4). The postures characterized by MAMMAL covered most of the previously identified individual behaviors of pigs[21]. (iii) Social behavior recognition, which is another attractive topic in the brain science field[16,31,34]. We focused on rule-based social behavior analysis that has been well-established in mice[16]. Due to the power of 3D surface distance computation enabled by PIG model reconstruction, MAMMAL could not only recognize both static and dynamic social behaviors (Fig. 2f, Supplementary Movie 5) but also distinguish part-level social contacts such as "Head-Head", "Head-Body" or "Head-Limb" (Fig. 2g, Supplementary Fig. 7). All the above analyses achieved by MAMMAL were benefited from not only our PIG model but also our well-designed motion capture algorithms.

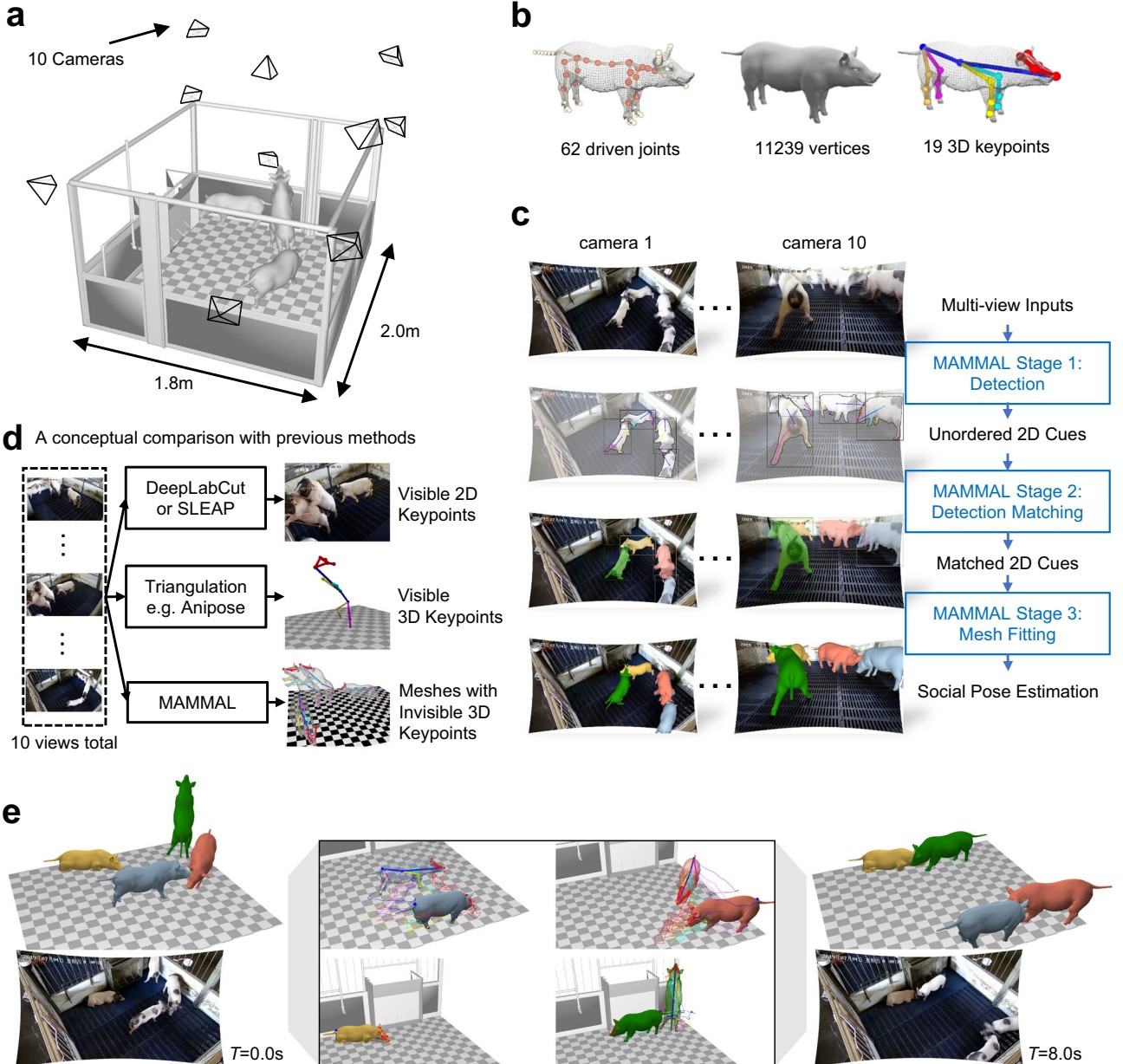

**Fig. 1 | MAMMAL presents a method for 3D surface motion capture of multiple animals. a** A pig motion capture setup. Ten cameras were set to shoot 4 freely behaving pigs in a cage. **b** The embedded joints (left, 62 joints), the surface (middle, 11239 vertices), and the regressed keypoints (right, 19 3D key points) in our PIG model. Among the 62 joints, we employed the 24 crucial joints labeled in light for pig body motion reconstruction while kept the rest fixed. **c** A representative overview of pig social motion reconstruction by the MAMMAL. **d** A visualized comparison with DeepLabCut/SLEAP (PMID: 35414125 [https://doi.org/10.1038/s41592-022-01443-0], 35379947 [https://doi.org/10.1038/s41592-022-01426-1]) or traditional triangulation-based methods like Anipose (PMID: 34592148 [https://doi.org/10.1016/j.celrep.2021.109730]). With the same 10-view input, our mesh fitting can predict the invisible 3D keypoints. **e** Qualitative results of MAMMAL for complex social behavior reconstruction.

## MAMMAL accurately tracks both visible and invisible keypoints

To verify the reliability of MAMMAL, a 3D evaluation dataset (so-called BamaPig3D) was collected from a 70-second video, which was manually labeled with 280 ground-truth (GT) pig instances containing 5320 3D keypoints, 2437 2D keypoints, and 2437 2D silhouettes (Supplementary Fig. 8). By evaluating on the BamaPig3D dataset, MAMMAL tracked all 19 pig keypoints accurately with an average error lower than 5.2 cm (Fig. 3a). The average error of all keypoints was 3.44 cm, which was lower than 5% pig body length. Specifically, MAMMAL not only precisely estimated body parts with sharp features and infrequent occlusions (such as eyes, ears, knees, elbows, mean error less than 3 cm) but also performed comparably well for terminal points with ambiguous features or frequent occlusions (such as nose, tail root,

paws, and feet, mean error less than 5 cm) (Fig. 3a). In fact, the detailed terminal point features are substantial for the understanding of abundant social signals for all animals including humans[47]. Moreover, MAMMAL performed equally well (mean error <7 cm, approximately 10% pig body length) in the estimation of invisible keypoints, which were keypoints visible to no more than one view, as in that of visible ones for most keypoints, except keypoints far from the body center such as nose, paws, and feet (Fig. 3b). To quantify the surface estimation accuracy, the pig meshes were rendered on each view and compared with manually labeled silhouettes, using the average of Intersection over Union (IoU). MAMMAL achieved an IoU of 0.77 using only predicted 2D keypoints for mesh fitting, and the predicted silhouettes further facilitated MAMMAL to achieve a higher IoU of 0.80

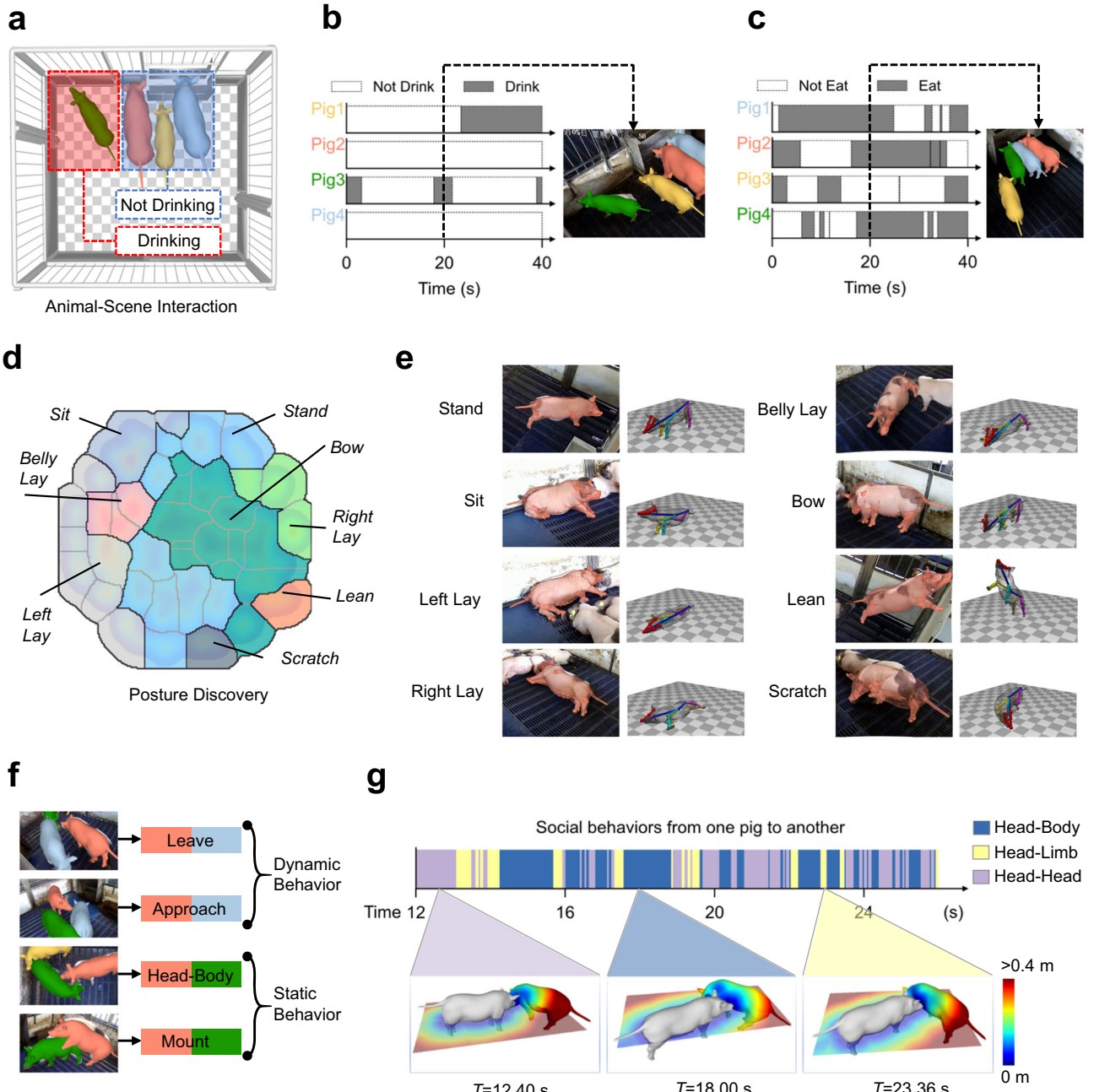

**Fig. 2 | MAMMAL enables quantitative analysis of rich pig behaviors.**
**a** MAMMAL can automatically identify drinking behavior during a feeding process. **b**, **c** Drinking (**b**) and eating (**c**) recognition by MAMMAL on two 40-second videos. Right, the 20th second for the pose estimation. **d** The density map of *t*-SNE space overlapped with isolated posture blocks obtained from the watershed transform of the density map. **e** An illustration of 8 distinct poses. For each case: left, raw image; middle, reconstructed mesh overlay; right, rendering of normalized pose by removing global transformation. **f** MAMMAL automatically identifies several defined social behaviors, and 2 dynamic behaviors and 2 static behaviors are shown here. **g** Social behaviors from one pig to another on a video clip. Representative 3D distance fields of three different social behaviors at three different timepoints are shown in subfigures. The surface model and distance fields help MAMMAL to distinguish "Head-Body", "Head-Head", and "Head-Limb" behaviors.

(Fig. 3c). Although only middle-sized pigs were labeled for training, MAMMAL was robust to different shapes of pigs and achieved similar error rates for new pig identities with different body sizes (Fig. 3d). Specifically, MAMMAL achieved an error rate of 2.3 ± 1.89 cm (mean ± SD) for one of the pigs in BamaPig2D dataset, 3.28 ± 2.15 cm for a moderate-sized pig, 4.27 ± 3.43 cm for a very fat pig with a large belly, and 3.78 ± 2.71 cm for a juvenile pig with small body size (Fig. 3e). The qualitative results further demonstrated the generalizability of MAMMAL to capture pigs with different ages, sizes, numbers or identities (Supplementary Movie 6), benefiting from the richness of BamaPig2D dataset and the flexible MAMMAL system.

## MAMMAL was more robust to camera numbers compared with triangulation

To show the superiority of MAMMAL over previous methods, we first compared MAMMAL Detection with SLEAP[31], which was trained on the BamaPig2D dataset as well. Our MAMMAL Detection outperformed both the bottom-up and top-down variants of SLEAP especially on side-

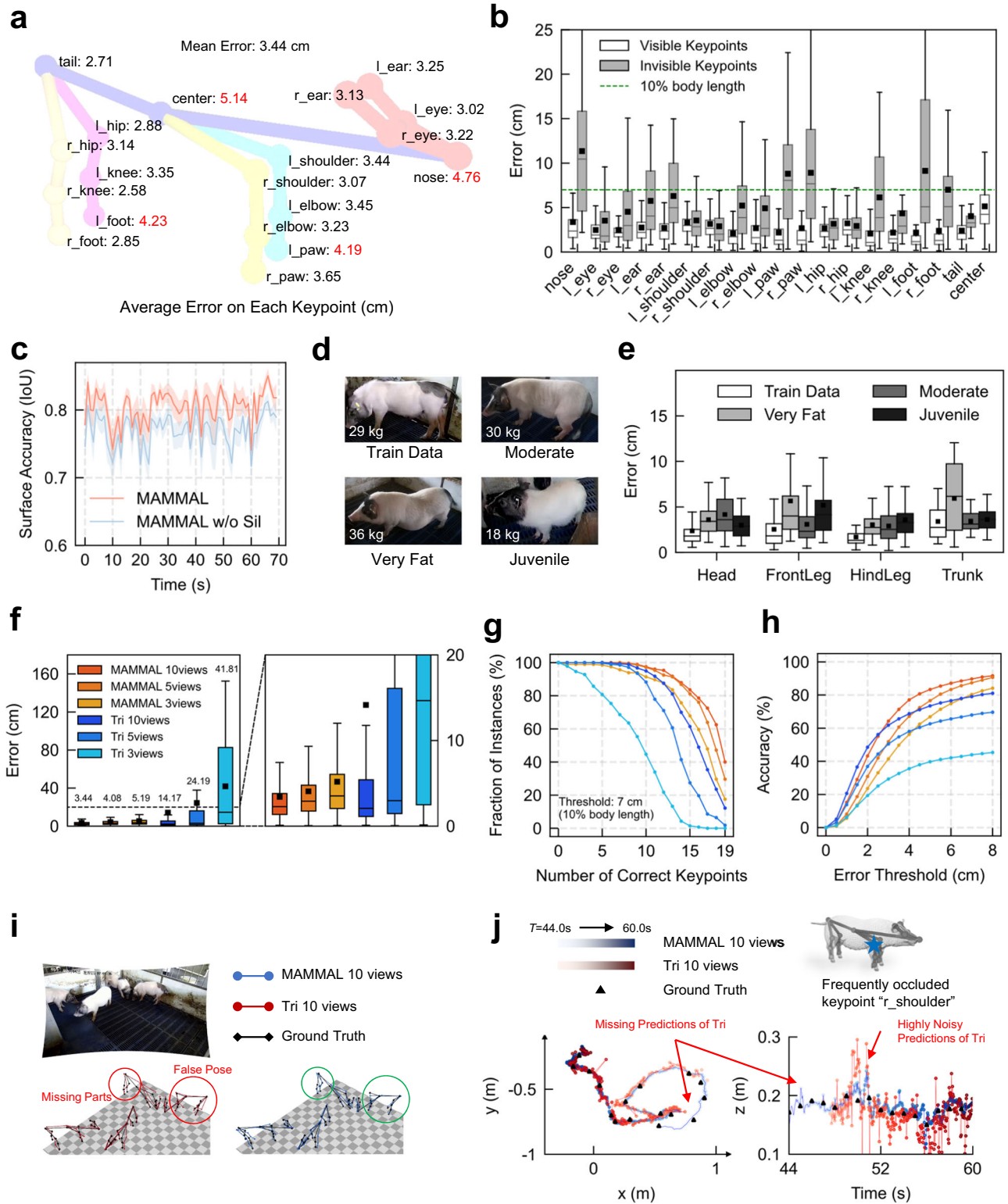

view images, demonstrating the effectiveness of resolution normalization to ensure equally good performance for different views (Supplementary Fig. 9). We then compared MAMMAL Mesh Fitting with traditional triangulation ('Tri' for short)[35–38] on the BamaPig3D dataset using different numbers of input views. Not surprisingly, MAMMAL stably reconstructed the 3D motions of pigs even with sparse view settings (3 views), while Tri barely functioned in fast motion and with occlusions (Supplementary Movie 7). Quantitatively, we compared the mean per-joint position error (MPJPE) and found that MAMMAL

achieved a much lower error than Tri with 10 views as input (Fig. 3f, MAMMAL 3.44 ± 3.99 cm, Tri 14.17 ± 32.02 cm, mean ± SD). Moreover, MAMMAL maintained a low error rate with sparse view settings (4.08 ± 4.45 cm for 5 views, 5.19 ± 6.10 cm for 3 views), while Tri yielded significantly larger mean error and greater variance (24.19 ± 39.73 cm for 5 views, 41.81 ± 43.23 cm for 3 views) due to incomplete or erroneous 3D pose estimation (Fig. 3g, Supplementary Movie 7). In addition, the number of keypoints accurately tracked by MAMMAL was significantly greater than that by Tri within a specific error threshold

**Fig. 3 | MAMMAL provides a robust solution for pose estimation compared to previous methods. a** Average 3D pose error of each keypoint (cm). Red indicates errors larger than 4.0 cm. Prefixes 'l_' and 'r_' indicate 'left' and 'right', respectively. **b** Box plot of MAMMAL reconstruction errors on both visible and invisible keypoints using 10 views (n = 4 animals, n = 70 timepoints). The dashed green line indicates an error of 7 cm (10% of the pig body length in BamaPig3D dataset). The 'center' part has no invisible keypoints. **c** Intersection over Union (IoU) versus timepoints on BamaPig3D dataset for surface estimation accuracy. 70 timepoints were used. At each timepoint, IoUs were averaged over all the labeled 2D instances of 10 views. 'MAMMAL w/o Sil' means without silhouette information during mesh fitting. Shadows, standard error mean (SEM). **d** An illustration of pigs with different weights and sizes for evaluation. Train Data, one of the pigs in BamaPig2D dataset. Other pigs are new identities, including 1) Moderate, a pig with similar body size to Train Data; 2) Very Fat, a pig with large belly; 3) Juvenile, a pig with very small body size. **e** Box plot of MAMMAL reconstruction errors on different pigs shown in **d**. n = 188, 188, 211 and 178 landmarks for 'Train Data', 'Very Fat', 'Moderate' and 'Juvenile' respectively. **f–h**, Box plot of 3D pose error of MAMMAL and triangulation (Tri) at different view configurations on the BamaPig3D dataset (n = 4 animals, n = 70 timepoints, n = 5320 landmarks) (**f**). Fraction of instances versus the number of correctly reconstructed keypoints of an instance. The error threshold used for determining a correct keypoint is 7 cm (**g**). Percentage of correctly tracked keypoints versus different thresholds (**h**). **i** A qualitative comparison between MAMMAL and Tri using 10 views. Tri produced missing parts and false poses especially for legs. **j** Coordinate curves of 'r_shoulder' keypoint of one pig on BamaPig3D dataset. Tri often yielded missing predictions or highly noisy predictions due to frequent occlusions. In **b**, **e** and **f**, black bar, median; box shoulders, interquartile range (IQR); whiskers, 1.5 times the IQR; black square dots, mean.

(7 cm) (Fig. 3g). Surprisingly, MAMMAL based on only 3 views was able to produce better performance than that of Tri based on 10 views, showing its view efficiency for pose estimation (Fig. 3g, h). Different from Tri, MAMMAL tracked the occluded points without missing or highly noisy predictions (Fig. 3i, j), demonstrating MAMMAL's ability to capture closer social interactions.

## Using MAMMAL to quantify the tail motion for pigs in different social hierarchies

To reconstruct the tail motions and quantify the motions in pigs in different social hierarchies[1,18,19], we recorded videos of dominant pigs (n = 4, with bigger body size) and subordinate pigs (n = 4, with smaller body size) during their feeding (Fig. 4a). Our quantitative analysis revealed that the dominant pigs monopolized the trough (Fig. 4b, c). We also found that the subordinate pigs usually touched the dominant pigs more frequently, while the dominant pigs focused on feeding instead of social interacting (Fig. 4d). To quantify tail movements, we added two keypoints on the pig tail (TailMid and TailTip) for 2D pose estimation and fit the tail motion with the full joint freedoms of the PIG model (Fig. 4e). The average error rate of pig tail was 6.05 cm, and the most flexible keypoint (TailTip) had the largest error rate due to the most severe motion blur (Fig. 4f). According to previous ethological study[3], passive hanging is more frequently observed in pigs that are exposed to an aversion situation, while loosely wagging is more related to positive emotions. In agreement with the previous report, we observed more loosely wagging but less passive hanging shown in the dominant pigs than in the subordinate pigs (Fig. 4g–j). The frequency of tail angle oscillations varies over time, which was similar to the limb movements of mouse grooming; therefore, we adopted a previously-reported parameter PSD (power spectral density)[40] that was employed to reflect limb movements of mouse grooming to measure the pig tail angle oscillations (Fig. 4g, h). Within a 10-second time window (Fig. 4h), the PSD value (10.24 $V^2 \cdot Hz^{-1}$) of the loosely wagging behavior peaked at 1.625 Hz, which was much higher than that of the passive hanging behavior (PSD = 0.45 $V^2 \cdot Hz^{-1}$ peaked at 0.125 Hz), indicating the effectiveness of using PSD to classify tail behaviors (Fig. 4i). By automatically determining loosely wagging behavior whose PSD is higher than 1.5 $V^2 \cdot Hz^{-1}$ across all the time windows, we found that the dominant pigs' tails oscillated significantly more than that of the subordinate pigs, which usually kept stationary (Fig. 4j, Supplementary Movie 8). Therefore, the detailed behavior recognition by MAMMAL is proven to be valid at least in the tail movements in different social hierarchies.

## MAMMAL is competitive with DANNCE-T for tracking single mouse extremities

Previously, DANNCE[40] achieved high accuracy for tracking 22 3D keypoints of single markerless mouse by pre-training on the million-scale Rat7M dataset and subsequently finetuning on 172 manually labeled frames. We compared MAMMAL with the temporal version of DANNCE, named DANNCE-T[48], on the 'markerless_mouse_1' sequence presented by DANNCE which was captured using 6 cameras surrounding an open field (Fig. 5a). We spent one day modifying the mesh model of a previously presented virtual mouse[49] for mesh fitting, which consisted of 140 driven joints and 14522 vertices, and we further defined 22 keypoints on it (Fig. 5b). For fair comparison, we trained HRNet using the same 172-frame training data to DANNCE-T by projecting original 3D keypoint labels to 2D. Consequently, MAMMAL successfully tracked the surface of the mouse and performed competitively to DANNCE-T for limb tracking (Fig. 5c, Supplementary Movie 9). For quantitative evaluation, we manually labeled 8 extremities (four paws, nose, two ear tips, tail tip) on another evenly distributed 50 frames on the 'markerless_mouse_1' sequence (Fig. 5d). When using total 6 views, MAMMAL achieved a lower error rate of 2.43 ± 1.69 mm (mean ± SD) than that of DANNCE-T (4.78 ± 7.15 mm, Fig. 5e). Except 'tail' point on which DANNCE-T had bad performance, the average error rate of MAMMAL (2.20 ± 1.25 mm) was still lower than that of DANNCE-T (2.71 ± 3.59 mm). Moreover, because DANNCE-T was directly trained in 3D volume space, its performance was sensitive to the camera setting (e.g., camera number) during inference. By testing on only 3 cameras, the performance of DANNCE-T degraded much faster than that of MAMMAL (Fig. 5e, f). Therefore, both qualitative and quantitative comparisons proved that MAMMAL is not only competitive with DANNCE-T for tracking mouse extremities but also more robust to different camera settings during inference.

## Using MAMMAL for social dog motion capture

To further demonstrate the generalizability of MAMMAL on social motion capture of other large-size mammals, we captured two male Beagle dogs in a laboratory environment using 10 GoPro cameras (Fig. 5g). Similar to the PIG model, we created a Beagle dog mesh model which consisted of 39 driven joints, 4653 vertices, and 29 keypoints (Fig. 5h). Following the pipeline of using MAMMAL for multiple animals and other animal species (Supplementary Fig. 10), we first labeled the 3D keypoints of 90 frames and then projected 3D keypoints to each view, resulting in 900 images with 2D keypoints labeling for training HRNet. We further labeled the segmentations using SimpleClick[50] for training PointRend. The mesh creation together with labeling took up 2 days. Finally, we reconstructed the dynamic social interaction of Beagle dogs using MAMMAL (Fig. 5i, Supplementary Movie 10). As a comparison, we trained VoxelPose[51], a strong volume-based baseline for multiple human 3D pose estimation, on the same 90-frame 3D labels. For a fair comparison, we labeled another 23 frames as test set and utilized the same HRNet results as inputs of VoxelPose during testing. By using 10-view inputs, MAMMAL achieved an average 3D keypoint error of 5.02 ± 3.22 cm (mean ± SD), which is lower than that of VoxelPose (5.93 ± 7.82 cm, Fig. 5j, k). By reducing the camera number from 10 to 6 and 4, MAMMAL achieved a slightly higher error rate of 5.44 ± 3.67 cm and 5.84 ± 4.06 cm (Fig. 5j, k). However, the error rates of VoxelPose increased more drastically to

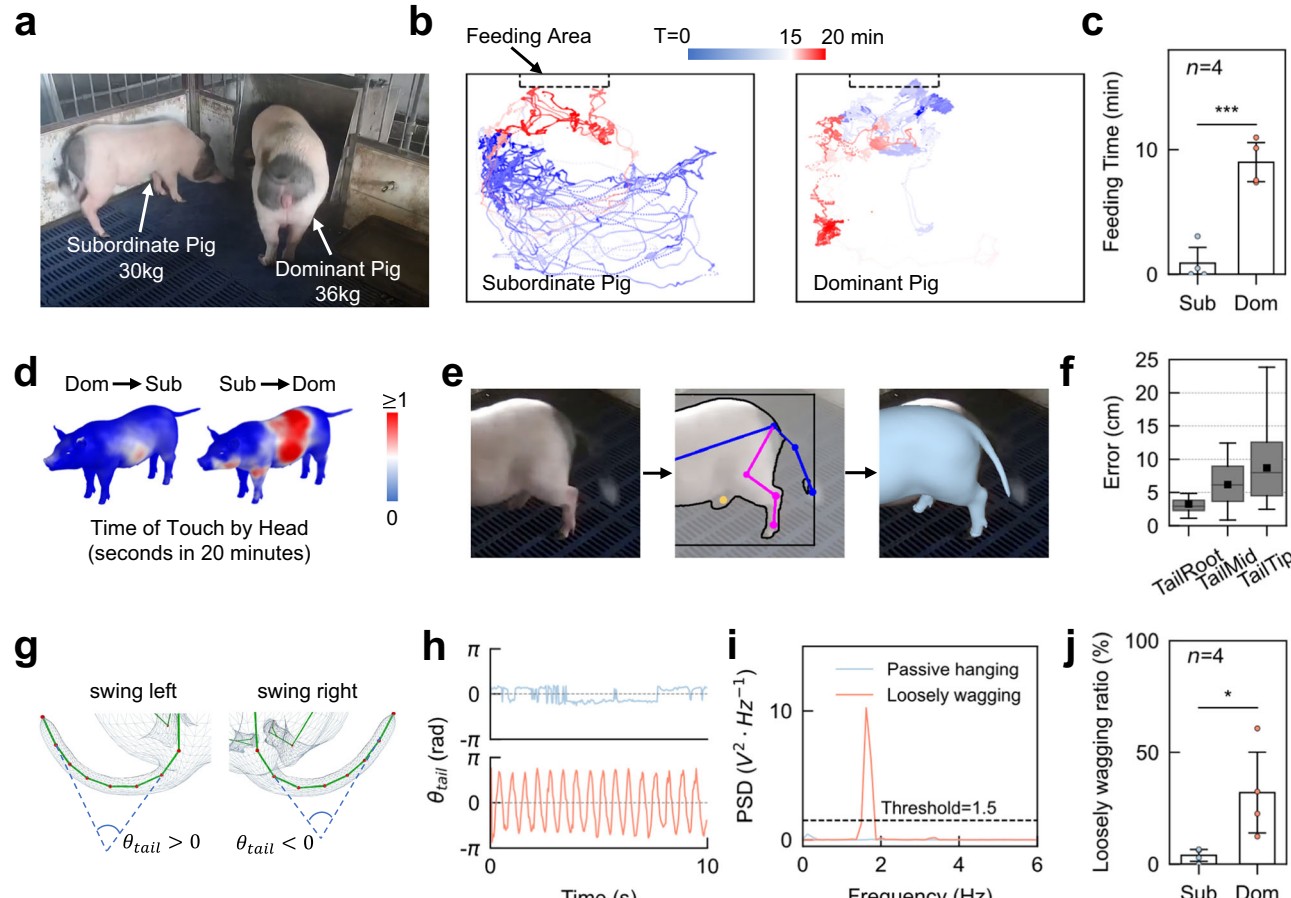

**Fig. 4 | MAMMAL quantifies the behavioral differences of two pigs in different social hierarchies. a** An illustration of a dominant pig (the bigger one) and a subordinate pig (the smaller one) at the timepoint of feeding. **b** Locomotion trajectories of the dominant and subordinate pigs during the 20-minute course of feeding. **c** Histogram of the feeding time for the dominant (Dom) and subordinate (Sub) pigs. Dominant pigs showed significantly longer feeding time than subordinate pigs ($p = 0.0004$, $n = 4$, two-sided independent samples $t$-test). **d** Heatmaps of the touch frequency by another pig's head. The left figure shows the frequency of the subordinate pig touched by the dominant pig's head, while the right figure shows the opposite. The heatmaps were averaged over four pairs. **e** An illustration of pig tail reconstruction. Given the original image (left), three tail keypoints were detected (middle), and the tail mesh of the PIG model were aligned (right). **f** Box plot of the MAMMAL reconstruction error on the three tail keypoints

($n = 2$ animals, $n = 20$ timepoints for each keypoint). Black bar, median; box shoulders, interquartile range (IQR); whiskers, 1.5 times the IQR; black square dots, mean. **g** An illustration of the tail angle computation. For tails swinging left, the tail angle $\theta_{tail}$ is positive, while the other side is negative. **h** Tail angle traces of two different tail behaviors in a 10-second time window. Blue, the passive hanging behavior. Red, the loosely wagging behavior. **i** Power spectral density (PSD) of tail angle traces in **h**. We used a PSD threshold of $1.5\ V^2 \cdot Hz^{-1}$ to determine whether a tail was wagging or hanging in a 10-second time window. **j** Histogram of the time ratio that a pig performed loosely wagging tail behavior during the 20-min feeding process. The loosely wagging ratio of dominant pigs were significantly larger than that of subordinate pigs ($p = 0.0367$, $n = 4$, two-sided independent samples $t$-test). In **c** and **j**, mean ± SD. Data were considered significant at $p < 0.05$ (*), with $p < 0.01$ (**), $p < 0.001$ (***).

7.17 ± 7.02 cm and 14.58 ± 19.79 cm (Fig. 5j, k), indicating that MAMMAL is more stable in a very sparse view setting for social dog motion capture without the need for training on the specific camera numbers. Taken together, we demonstrate that MAMMAL can be applied for tracking social motions in other large-size mammals in an efficient and accurate manner.

## Discussion

Overall, MAMMAL is the first described method that enables surface motion captures of multiple freely moving animals in a noninvasive manner in their living environments. The substantial advantages of MAMMAL over previous methods are in dealing with invisible key points and severe occlusions, especially for large-size animals. Although MAMMAL could not run in real-time currently, it enables various quantitative behavioral analyses, which may inspire studies in disease modeling, drug evaluation, brain circuit function characterization, etc. In addition, monitoring health condition of large-size mammals, such as pigs, will be valuable for the improvement of domestic animal red meat production and the prosperity of the

industries. For generalizing the usages, MAMMAL is also proven to be effective in tracking mouse and Beagle dogs without a million-scale 3D dataset for training.

Recently, as the articulated mesh models have achieved great success in human behavior modeling[52,53] and clinical research[54], their applications in animal behavior capture are in the ascendant[55-63]. Although keypoints can be used for several behavioral analyses, they are sparse and lack the ability to measure dense surface contact between animals/agents. In contrast, articulated meshes are very valuable to encapsulating anatomical priors of animals[49,58], simplifying the modeling of surface-to-surface contact, and serving as a medium to fuse multi-modal data like point clouds[55]. These advantages encouraged us to handle the challenges of social animal motion tracking using articulated mesh models.

At the current stage, our paper is not only a proof-of-principle paper but also a toolbox with basic requirements. For others to easily use MAMMAL as a toolbox, we made all our codes open-sourced, included instructions in Readme files of every code, summarized how to apply MAMMAL to multiple animals and other animal species in

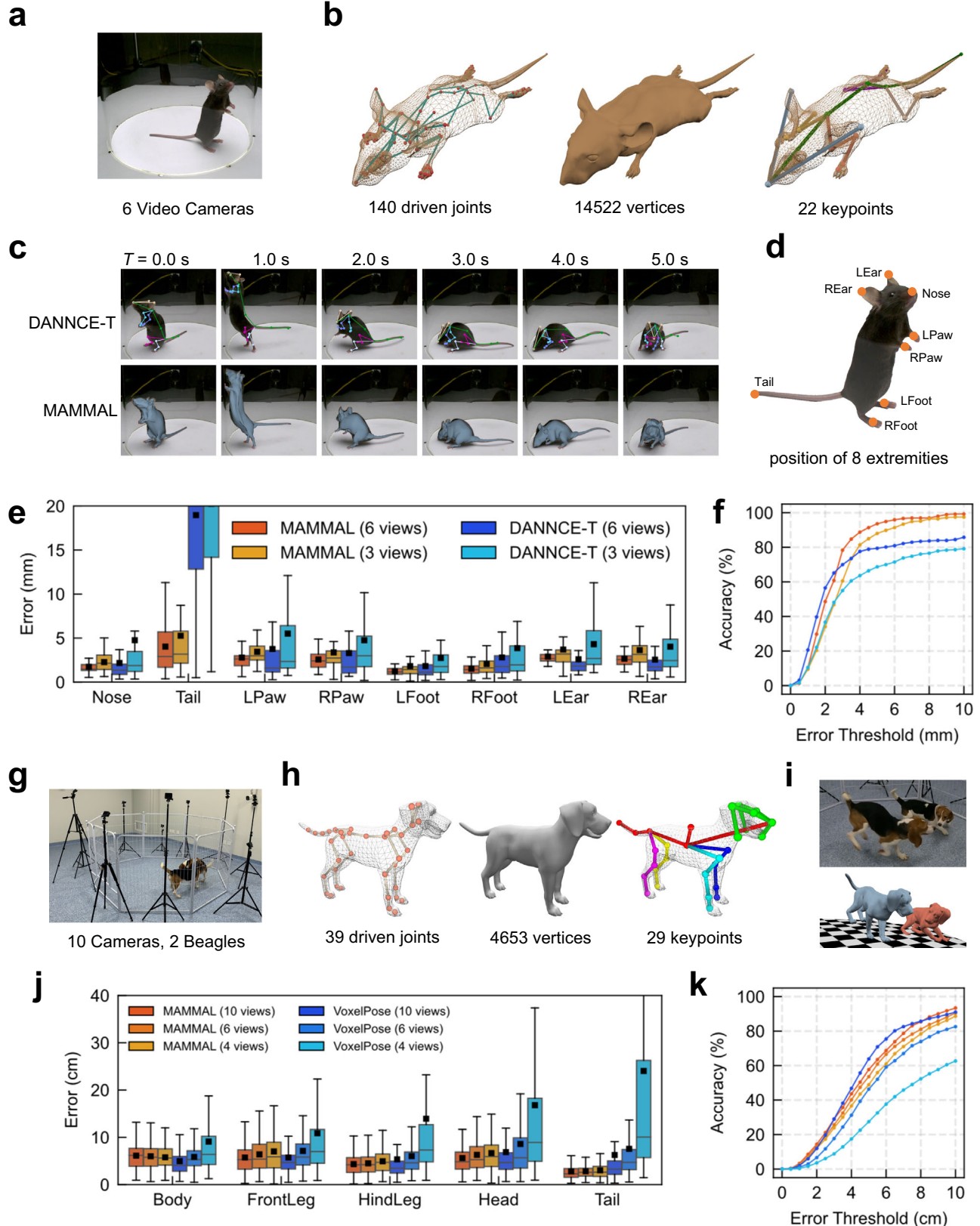

more user-friendly. Acceleration techniques[65,66] could be further applied to make MAMMAL faster for real-time applications. In the future, we expect to build a linear blend shape model for pigs similar to SMPL[52] for humans or SMAL[63] for quadrupeds.

Supplementary Fig. 10, and ensure that the computational processing of MAMMAL can be handled by an undergraduate student with C ++/ Python coding experiences. Several aspects could be considered for improving MAMMAL. For example, merging widely used 2D pose estimation[31,34], segmentation, identification, and behavior classification[64] into MAMMAL as an end-to-end system could make it

Overall, MAMMAL provides a tool for analyzing the surface motions of single or socially interacting mammals such as pigs, mice,

**Fig. 5 | Using MAMMAL for mouse and dog motion capture. a** An illustration of the video data used for reconstruction. Six views were used. **b** The mouse mesh model used by MAMMAL. It contains 140 driven joints (left), 14522 vertices (middle), and 22 keypoints (right). **c** Visualized comparison between DANNCE-T and MAMMAL. **d** The position of 8 extremities of mouse labeled for quantitative comparison. **e, f** Box plot of reconstruction errors of different methods on the 8 extremities. Total 50 evenly distributed frames were manually labeled for evaluation (**e**). Percentage of correctly tracked keypoints versus different thresholds (**f**). **g** An illustration of the laboratory environment of Beagle dog social motion

capture. Ten GoPro cameras were mounted round the experiment area where two beagle dogs freely moved. **h** The dog mesh model used by MAMMAL. It contains 39 driven joints (left), 4653 vertices (middle), and 29 keypoints (right). **i** A 3D rendering of reconstructed dog poses. **j, k** Box plot of reconstruction errors of MAMMAL and VoxelPose on different view numbers ($n = 2$ animals, $n = 91, 366, 368, 351$ and $138$ landmarks for Body, FrontLeg, HindLeg, Head and Tail respectively) (**j**). Percentage of correctly tracked keypoints versus different thresholds (**k**). In **e** and **j**, black bar, median; box shoulders, interquartile range (IQR); whiskers, 1.5 times the IQR; black square dots, mean.

and Beagle dogs with the help of an articulated mesh model. The concept and algorithms of MAMMAL we described here may apply to other animal detailed behavior analysis and health condition monitoring both in agriculture (e.g., cattle, sheep) and in life science (e.g., rats, macaques).

## Methods

### Data collection and animals

We set 10 HIKVISION network cameras (DS-2DE2204IW-DE3/W series) around a 2.0 m × 1.8 m cage to achieve 1920 × 1080 @ 25FPS multi-view videos (256 GB SD card/camera for data storage). Three-day videos of four male Bama pigs (17 weeks of age) were referred to as Seq1. We sampled 3340 images from Seq1 to create the BamaPig2D dataset, and a 1750-frame video clip from Seq1 to create the BamaPig3D dataset. The annotations of BamaPig2D dataset followed the instruction of COCO[43], which is a commonly used annotation protocol in computer vision community. The BamaPig3D dataset was manually labeled with the Python package LabelMe v4.5.7. For fair evaluation, the BamaPig3D and BamaPig2D datasets had no timepoint overlap. For qualitative behavior analysis, we further captured seven-day videos of the same four pigs at 30 weeks of age (referred to as Seq2, 5 views) and one-day video of four different pigs (referred to as Seq3, 8 views).

### Camera calibration

To make the camera calibration more accessible, a calibration pipeline was employed to align manually labeled scene points with actual captured images (Supplementary Fig. 1). The intrinsic calibration was performed only once for all cameras, and the extrinsic calibration was performed without the structure from motion (SfM) that was utilized by traditional methods[37,38]. To make intrinsic calibration easier, all the cameras were set to the same maximum focal length (Seq1, Seq2 with the slightest distortion, recommended) or the same minimum focal length (Seq3 with the most severe distortion, for a larger field of view). We calibrated both types of distortion parameters in advance using the chessboard and OpenCV calibration toolbox[67] and chose suitable distortion parameters case by case. For extrinsic calibration, we manually labeled easily recognized scene points according to the geometric features of the captive cage and calibrated the videotaped images to the points. Afterwards, the projective n-points (PnP) algorithm was applied to compute the R and t of each view using OpenCV.

### PIG model formulation

PIG is an articulated mesh model with $N^V = 11239$ vertices and $N^J = 62$ embedded joints. The model contains template surface points $\bar{T} \in \mathbb{R}^{N^V \times 3}$ and template joint locations $\bar{J} \in \mathbb{R}^{N^J \times 3}$ in the rest pose, together with sparse skinning weights $W$. The i-th joint is attached with 3 degrees of freedom (DOF) rotation $\phi_i \in \mathfrak{so}(3)$ in Axis-Angle format. Among the 62 joints, 24 critical ones were employed for pig locomotion and social interaction prediction with the rest of joints were fixed. The posed surface points $V$ are determined by linear blend skinning (LBS) process as

$$V = M(\Theta) = M(\theta, s, \boldsymbol{r}, \boldsymbol{t}) = s \cdot R(\boldsymbol{r}) \cdot LBS(\theta; \bar{T}, \bar{J}, W) + \boldsymbol{t} \quad (1)$$

where $\Theta$ represents the full parameter set, $\theta \in \mathbb{R}^{N^O \times 3}$ is the stack of $\phi_i (i \in \{1, 2, \ldots, N^O\})$, $s$ is the global scale, $\boldsymbol{r} = [r_z, r_y, r_x]^T$ is the global rotation represented as Euler angles, $R(\boldsymbol{r})$ is the rotation matrix yielded by $\boldsymbol{r}$, and $\boldsymbol{t} = [t_x, t_y, t_z]^T$ is the global translation. A sparse skeleton regressor $J \in \mathbb{R}^{N^K \times N^V}$ is additionally designed to regress 3D keypoint positions $X \in \mathbb{R}^{N^K \times 3}$ from mesh vertices $V$ as $X = JV$, where $N^K = 19$ is the number of keypoints. Note that, the PIG model was built manually by an artist in one day. For an animal species that does not have the 3D mesh, we recommend to follow the process as previously described[49] to create a customized 3D mesh model. If the mesh model does not fit the shape of the animal species well, we recommend to tune the mesh model according to the captured images using MAYA software or automatically deform the mesh vertices using non-rigid deform algorithms according to the captured silhouettes.

### Training 2D detection networks

MAMMAL employed PointRend[45] for pig silhouette detection and HRNet[46] for pig pose estimation. Both networks were trained on the BamaPig2D dataset, with 90% images (3008 images, 10356 instances) for training and 10% (332 images, 1148 instances) for testing. PointRend was trained for 270k iterations with 8 images per batch, and the whole training procedure took 5 days. Finally, PointRend achieved an average precision (AP) of 0.869 for bounding box detection and 0.868 for silhouette segmentation. HRNet was trained for 120 epochs with batch size 16 and the training procedure took 32 hours, resulting in an AP of 0.633. Both networks were trained and tested on Ubuntu LTS 18.04 system with a single NVIDIA RTX 2080Ti GPU.

### MAMMAL Stage 1: Detection

Given that $N^C$ synchronized multi-view images $I = \{I^c\}_{c \in \{1, 2, \ldots, N^C\}}$, MAMMAL first performed 2D detection in a top-down manner. For view $c$, PointRend was adopted to generate bounding boxes $B_q^c (1 \leq q \leq N^{B_c})$ and silhouettes $S_q^c$ of all visible individuals. Here, $N^{B_c}$ was the number of detected pigs on view $c$, $S_q^c$ was 1920 × 1080 binary mask image where '1' means foreground part and '0' means background part. Without ambiguity, we could refer to a pig by its bounding box $B_q^c$. In fact, $N^{B_c}$ may be different across views because the occlusion relationships of pigs were different by observing from different views. Afterwards, we cropped out each pig image $I_q^c$ according to the bounding box $B_q^c$ for $q$-th pig on view $c$ and normalized its resolution to 384 × 384. Then the normalized pig image was fed to HRNet to obtain the prediction of $N^K$ keypoints. The keypoints for $q$-th pig on view $c$ was $Y_q^c = \{y_{q,m}^c \in \mathbb{R}^2, \sigma_{q,m}^c\}_{m \in \{1, 2, \ldots, N^K\}}$. Here, $y_{q,m}^c$ was the 2D coordinate of the $m$-th keypoint, $\sigma_m^c$ was the visibility of the $y_{q,m}^c$ ($\sigma = 0$ meant invisible, $0 < \sigma \leq 1$ meant visible confidence). Before Detection Matching of MAMMAL, the 2D cues $Y_q^c$ and $S_q^c$ were unordered, which meant that for any pig on view $c_1$, we did not know which pig on view $c_2$ corresponded to it ($c_1, c_2 \in \{1, 2, \ldots, N^C\}, c_1 \neq c_2$). Therefore, we must match the 2D cues both spatially and temporally before predicting the 3D pig postures.

## MAMMAL Stage 2: Detection Matching

At the first frame of a sequence ($T = 0$), there was no available temporal information, therefore we must match unordered 2D cues from different views using only spatial information. To this end, MAMMAL first built a 3D association graph by viewing all the detected instances as graph nodes. Graph edges were only defined to connect nodes belonging to different views, resulting in a sparse multipartite graph. The average epipolar distance of visible keypoints across views were calculated as edge weights. We partitioned this graph by modifying a maximal clique enumeration (MCE) method[68] to cluster all unordered 2D cues into $N^p + 1$ groups, with $N^p$ groups corresponding to $N^p$ pig identities and one more group containing useless wrong 2D cues.

The time complexity of our graph partitioning algorithm was $O((N^C)^{N^P})$, which meant that if the pig number $N^p$ was fixed, the running time was polynomial to the view number $N^C$. At the subsequent frames ($T > 1$), we projected the 3D keypoints regressed from the PIG model at $T$-1 (0.04 s prior to $T$) on each view and solved the per-view tracking problem using Kuhn-Munkres algorithm according to the average Euclidean distance of keypoints. Consequently, all the 2D cues at $T > 0$ were divided into $N^p + 1$ groups too. Thus, for each subject, the matched 2D cues on view $c$ were $Y = \{y_m^c \in \mathbb{R}^2, \sigma_m^c\}_{m \in \{1,2,...,N^K\}}$ and $S^c$, where $y_m^c$ was the 2D coordinate of the m-th keypoint, $\sigma_m^c$ was the visibility of the m-th keypoint, and $S^c$ was the silhouette.

## MAMMAL Stage 3: Mesh Fitting

At $T = 0$, MAMMAL first retrieved a pose hypothesis referred to as 'anchor pose' from a pose library containing $N^L$ predefined poses. Specifically, for the p-th ($p \in \{1,2,...,N^L\}$) pose in the library, we divided the full pose parameters into a constant part $\Theta_p^l$ and a variable part $\Theta_p^g$. In detail, $\Theta_p^l = \{\theta, t_z, r_x, r_y\}_p$ encodes pose information fixed during pose retrieval and $\Theta_p^g = \{t_x, t_y, r_z, s\}_p$ encodes global transformation that varies among different subjects. Note that $\{t_z, r_x, r_y\}_p$ in $\Theta_p^l$ carries important prior information on how pigs naturally interact with the scene, preventing impossible initial pig postures such as 'flying' or 'underground'. The retrieval score for the p-th pose was defined as the energy function $E(\Theta_p^g) = w_{2D} E_{2D}(\Theta_p^g) + w_{sil} E_{sil}(\Theta_p^g)$. $E_{2D}(\Theta_p^g)$ is the 2D energy term penalizing the accumulated Euclidean distance error between the multi-view 2D keypoints and the projections of p-th subject's 3D keypoints. $E_{sil}(\Theta_p^g)$ is the silhouette energy term measuring the Chamfer distance between the projected mesh silhouette vertices $\hat{S}^c(\Theta_p^g)$ and the distance transform of the detected silhouette $S^c$. For time efficiency, we first minimized $E_{2D}(\Theta_p^g)$ and then searched the optimal $p^*$ that minimizes $E(\Theta_p^g)$. In our implementation, $w_{2D} = 1, w_{mask} = 20$, and tail vertices were ignored. The retrieved anchor poses acted as initial poses for $T = 0$. At $T > 0$, we simply inherited pig poses from $T$-1 as initial poses. After obtaining initial pig poses at each time $T$, we defined the full minimization objective function $E(\Theta_T)$ as the weighted sum of several independent energy terms. Without ambiguity, subscript $T$ is here omitted except for the temporal term. $E(\Theta)$ is defined as

$$E(\Theta) = w_{2D} E_{2D} + w_{sil} E_{sil} + w_{temp} E_{temp} + w_{reg} E_{reg} + w_{anchor} E_{anchor} + w_{floor} E_{floor}$$

Among all the energy terms, $E_{2D}$ fits pose parameter space $\Theta$ to match 2D keypoints on each view, which is the same as $E_{2D}$ for anchor pose retrieval. Concretely,

$$E_{2D}(\Theta) = \sum_{c=1, m=1}^{N^c, N^K} v_m^c |\pi^c(X_m(\Theta)) - y_m^c|_2^2 \qquad (2)$$

$\pi^c(\cdot)$ is the camera projection function for view $c$, which projects a 3D point in the global coordinate system to the local image plane, and $X_m(\Theta)$ is the m-th 3D keypoint position regressed from the PIG model driven by $\Theta$. The construction of the silhouette term $E_{sil}(\Theta)$ is a differentiable rendering process, where all $N^p$ pigs are rendered simultaneously to each view, and the parameters of different individuals are optimized together. For a specific pig, we first determined the visibility of all its vertices by comparing its own rendered depth map to the depth map with all pigs. Then for visible vertices, we construct $E_{sil}$ as

$$E_{sil}(\Theta) = \sum_{q \in B} [SDF_P(\pi^c(V_q(\Theta))) - SDF_D(\pi^c(V_q(\Theta)))]^2 \qquad (3)$$

where $B$ is the set of visible vertex indices, $V_q$ is the q-th vertex, $SDF_D$ is the signed distance function (SDF) calculated from the 2D silhouette generated by PointRend, and $SDF_P$ is SDF calculated from the rendered silhouette. Here, SDF is an extension to distance transform where all pixels range in $[-1,1]$ with 0 corresponding to the silhouette. The temporal term $E_{temp}$ regularizes current subject keypoints $X_m(\Theta)$ with keypoints at $T - 1$ by

$$E_{temp}(\Theta_T) = \sum_{m=1}^{N^K} |X_m(\Theta_T) - X_m(\Theta_{T-1})|_2^2 \qquad (4)$$

The regularization term $E_{reg}(\Theta) = |\theta|_2^2$ constrains the joint rotations to be small. At $T = 0$, term $E_{anchor}$ uses the anchor pose as a strong prior on invisible keypoints. $E_{anchor}$ is written as

$$E_{anchor} = |\Theta^l - \Theta_{p^*}^l|_2^2 + \lambda_h \sum_{m=1}^{N_K} |X_m^{(z)} - X_{m,p^*}^{(z)}|_2^2 \qquad (5)$$

where the first part constrains each pose parameter to be close to anchor pose $\Theta_{p^*}^l$, $X_m^{(z)}$ is the z coordinate of keypoint $X_m$, and $\lambda_h = 25$ in our implementation. The floor term $E_{floor}$ is used to force all keypoints to be above the floor and is written as $E_{floor} = \sum_{m=1}^{N_K}(\text{Ramp}(-X_m^{(z)}))^2$, where Ramp is the ramp function with a slope equal to 1. A typical set of term weights was $w_{2D} = 1$, $w_{sil} = 5 \times 10^{-5}$, $w_{reg} = 0.01$, and $w_{floor} = 100$. At $T = 0$, $w_{temp} = 0$ and $w_{anchor} = 0.01$; at $T > 0$, $w_{temp} = 1$ and $w_{anchor} = 0$. The optimization exploits the Levenberg-Marquardt algorithm with less than 60 iterations for initialization and less than 15 iterations for tracking. In practice, we set $w_{sil} = 0$ during the first 5 iterations.

### Animal-Scene interaction measurement

Two scene-related behaviors were defined for automatic recognition: drinking and feeding. MAMMAL recognized these behaviors by comparing the 3D coordinates of subject noses to scene features. Specifically, drinking was identified using the Euclidean distance between the reconstructed pig nose and the water tap position. The threshold in our case is 0.12 m. Eating was identified by measuring whether the nose is in a $0.62m \times 0.7m \times 0.2m (x \times y \times z)$ cube space corresponding to the feeding area. Video clips from Seq1 and Seq2 were utilized for demonstrating eating and drinking recognition, respectively (see also Supplementary Movie 3).

### Posture discovery

To discover distinct individual postures, we collected 44 small motion clips of 4 individual pigs, with each clip lasting for 1 to 10 s, resulting in 20819 postures in total. All these poses were first normalized by removing $\Theta^g$. Then, key parameters representing a pig posture were stacked into a 178 dims data vector, including the keypoint positions ($3 \times 19 = 57$ dims), the keypoint velocities ($3 \times 17 = 51$ dims without ears), the height of tail and center (2 dims), the angles of body pitch and roll (2 dims), and the Axis-Angle rotations of some joints ($3 \times 22 = 66$ dims). Next, the 178 dims data vector was reduced to a 16-

dim feature vector using principal component analysis (PCA), where 16 dims explain 97.3% of the data variance. *t*-SNE was then applied to create a 2-dim embedding of all the feature vectors, where the perplexity of *t*-SNE is 80. Before density estimation, both dimensions of the embedding vectors were normalized to the range [0.05,0.95]. The density map was obtained using Gaussian kernel density estimation with bandwidth 0.03 and further normalized to [0,1]. Finally, the watershed algorithm was used on the reverse of the density map to obtain clustered posture blocks (Fig. 2d). By manually checking the poses at the local density peaks of each block, we identified 8 distinct postures (Supplementary Movie 4). Individual behavior clustering algorithms were implemented using Python 3.7.9, numpy 1.17.5, scipy 1.6.0, sklearn 0.24.1 for *t*-SNE and density estimation, and skimage 0.18.1 for the watershed algorithm.

## Social behavior recognition

We defined seven dyadic social behaviors including two dynamic behaviors (approach and leave) and five static behaviors (head-head, head-body, head-limb, head-tail and mount) (Fig. 2f, g, Supplementary Movie 5). For two pigs *A* and *B* engaged in a social behavior in which the pig *A* was the active one, we first found a vertex $v_B^*$ on pig *B* whose distance to pig *A*'s head was minimum. Here the minimum distance was recorded as $d_h(A,B)$ and the body part of $v_B^*$ was denoted as $P(v_B^*)$ (Supplementary Fig. 7). To reduce the computational burden, we evenly down-sampled 546 vertices V' from the original 11239 vertices V and used V' for distance calculation. In addition, the top view overlay $o(A,B)$ was computed by first projecting $V_A'$ and $V_B'$ to the $xOy$ plane as $V_A'^{(xy)}$ and $V_B'^{(xy)}$ and then computing the Intersection over Union (IoU) of the convex hulls of $V_A'^{(xy)}$ and $V_B'^{(xy)}$. The body pitch angle of pig *A* was denoted as $r_y(A)$. With the above prepared data, the dyadic social behaviors at time *T* were identified in time window $[T-W,T+W]$ using the following equations:

$$
\begin{aligned}
\text{Approach}: \quad & d_h(A,B)^{T-W}>0.2, d_h(A,B)^{T+W}<0.05,\\
\text{Leave}: \quad & d_h(A,B)^{T-W}<0.05, d_h(A,B)^{T+W}>0.2,\\
\text{Head-Head}: \quad & d_h(A,B)^T<0.05, P(v_B^*)=head,\\
\text{Head-Body}: \quad & d_h(A,B)^T<0.05, P(v_B^*)=body,\\
\text{Head-Limb}: \quad & d_h(A,B)^T<0.05, P(v_B^*)=limb,\\
\text{Head-Tail}: \quad & d_h(A,B)^T<0.05, P(v_B^*)=tail,\\
\text{Mount}: \quad & r_y(A)>20^\circ, o(A,B)>0.15,
\end{aligned}
\tag{6}
$$

where $W=12$ typically. We tested the algorithm qualitatively on a 40-second video extracted from Seq3 and achieved accurate and robust dyadic social behavior recognition (see Supplementary Movie 5).

## Triangulation

Typically, there are two types of triangulation: direct linear transformation (DLT)[36,39,40,67] and maximum likelihood estimation (MLE) method[35–38,69]. Although these two methods often result in similar results, MLE is usually more robust because it minimizes the per-view projection error (geometric errors) in an iterative way and its results have clear geometric meaning. Therefore, we implemented MLE in all our experiments.

## SLEAP training and inference

We trained SLEAP v1.2.6 on BamaPig2D dataset. For the top-down variant of SLEAP, we set 'Sigma for Centroids' to 8.00 and 'Sigma for Nodes' to 4.00. We used Unet with 'max-stride' set to 64 as centroid model, and Unet with 'max-stride' set to 32 as centered instance model. Other parameters were set to default ones. The training process took half a day on Windows 10 with single NVIDIA RTX 3090Ti GPU (24GB). During inference on the videos of BamaPig3D dataset, 'flow' tracker was applied with 'similarity method' set to 'instance' and 'matching

method' set to 'greedy'. For bottom-up variant of SLEAP, we utilized Unet with 'max-stride' set to 64 as model. We also applied scale and uniform noise augmentation during training. During inference, we set the tracker setting same to the top-down variant of SLEAP.

## Experiments on different views

To fairly perform experiments between MAMMAL and triangulation for pose estimation, we utilized the same 2D cue matching results obtained by MAMMAL running on 10 views. For 5 view experiments, we utilized the top five camera views. For 3 view experiments, we utilized 3 top corner cameras to ensure good view distribution. All quantitative experiments were performed without temporal smoothing.

## Experiments on pig sizes

We first chose one of the four pigs in BamaPig2D dataset as the baseline pig size, whose weight was 29 kg ('Train Data' in Fig. 3d). We additionally captured two groups of pigs. One group consisted of moderate-sized pigs and another one consisted of very fat pigs, both of which was labeled with 12 frames of 19 3D keypoints; the other group consisted of four juvenile pigs, one of which was labeled with 12 frames of 19 3D keypoints for evaluation (Fig. 3d).

## Pig tail reconstruction and analysis

To capture the motion of pig tails, we additionally defined two keypoints on the pig tail. One is 'TailMid', which lies on the middle of pig tail. The other is 'TailTip', which lies on the tip of pig tail. We labeled 48 images with tail keypoints in BamaPig2D, and another 675 images with tail keypoints from newly captured sequences used for social rank analysis. We merged newly labeled data to BamaPig2D, resulting in an extended dataset BamaPig2D_Ext, and trained HRNet on it from scratch for 240 epochs. During mesh fitting, we used the full degrees of freedom of 62 joints, and added an energy term $E_{3D}$ to penalize the distance between regressed keypoints and triangulated keypoints. For faster inference, we set $w_{sil}=0$ to disable silhouette loss and set optimization iterations to 3 during tracking. To compute tail angles, we named the 8 tail joints of the PIG model as tail_1 to tail_8 (from tail root to tail tip). We first defined vector $\vec{a}$ as the 3D unit vector pointing from tail_1 to tail_2, and vector $\vec{b}$ as the 3D unit vector pointing from tail_7 to tail_8. We then calculated the tail angle as $\theta_{tail}=|\arccos(\vec{a}\cdot\vec{b})|$ sgn, where sgn is the sign of tail angle. To determine sgn, we defined a plane in 3D space determined by tail_1, tail_2 and pig body center, and judged whether tail_8 was on the left side of the plane (sgn > 0) or the right side (sgn < 0). The power spectral density (PSD) was calculated using the welch function in scipy 1.6.0 with sampling frequency as 25 and length of segment (nperseg parameter) as 200. We reported PSD in linear units.

## Mouse data preparation

As there was no available mouse articulated mesh model, we extracted and modified a mesh model from a previously proposed virtual mouse model[49]. Specifically, we exported the vertices, skinning weights and embedded joints from the original Blender file, and manually removed the whiskers using Blender and Meshlab. Finally, we obtained an articulated mesh model with 140 driven joints and 14522 vertices. Then we manually defined 22 keypoints on it, which corresponded to the body parts similar to what DANNCE[40] tracked. In total, the mesh preparation took 1 day. DANNCE proposed two sequences "markerless_mouse_1" and "markerless_mouse_2", each of which captured a freely moving mouse in an open field using 6 cameras for 18,000 frames in 1152 × 1024 @ 100 FPS. DANNCE annotated 172 frames in total on the two sequences for training. We projected the labeled 3D keypoints of these 172 frames to each view and obtained 1032 images with ground truth 2D keypoints to train HRNet. To reduce the burden of silhouette labeling, we applied the off-the-shelf segmentation toolbox SimpleClick[50] to automatically segment mouse using predicted 2D

keypoints as guidance. We only tested MAMMAL on "markerless_mouse_1", and manually labeled the 3D positions of 8 extremities on evenly distributed 50 frames of the sequence for quantitative evaluation.

## Mouse mesh fitting with volume preserving bones

Because the surface of mouse could stretch and deform larger than pigs in different poses, we made two adaptations of the mesh fitting pipeline. First, we added bone length parameters to account for body stretch. Second, we added volume preserving constraints[49] to account for belly deformations. These two adaptations made the results aligned to mouse silhouettes better (Supplementary Movie 9). Currently, the mouse mesh fitting was implemented in Python based on the same pipeline to C++ version of MAMMAL.

## DANNCE-T for less cameras

We used the officially released model trained on "markerless_mouse_1" and "markerless_mouse_2" for all the testing. Because such volume-based method requires the input camera number to be the same to the training one during inference, we copied the first three views twice as inputs to obtain the results on 3 views. We used the same three views for MAMMAL reconstruction for a fair comparison.

## Experiments on Beagle dogs

We recorded multiple-view videos of two socially interacting Beagle dogs in laboratory environment using 10 GoPro HERO11 Black cameras in 1920 × 1080 @ 120 FPS. Then, we purchased a Beagle model online (https://sketchfab.com/) as the basic articulated mesh model, and made some modifications (e.g., removed whiskers and useless bones) to fit our requirements. Then we defined 29 keypoints on it. The mesh preparation took us 1 day in total. To minimize the workload in data annotation, we directly annotated 3D keypoints of each dog for 90 frames, and projected them to each view to obtain ground truth 2D keypoints. The keypoint labeling took up 7 hours. Afterwards, with the assistance of SimpleClick[50], the silhouette labeling of 900 images took up 10 h. Then, the training process was similar to that of pigs, which took up 1.5 days. The dog matching and mesh fitting procedures were the same to pigs. We configured VoxelPose[51] with [2400, 2400, 2000] space size and [300, 0, -200] space center according to the coordinate system of our Beagle dog data. The initial cube size was [48, 48, 12]. We trained VoxelPose on the same 90-frame training data for 190 epochs with batchsize set to 4. The optimizer was Adam with learning rate 1e-4. The training and testing procedures of VoxelPose employed the 2D outputs of the same pretrained HRNet to that of MAMMAL. The whole training took 18 hours on Ubuntu 20.04 system equipped with single NVIDIA Geforce RTX 3090 GPU. To test VoxelPose on 6 views, we used camera [1,3,4,5,8,9] (0-indexed). To test VoxelPose on 4 views, we used camera [1,3,8,9] (0-indexed). We used the same cameras for MAMMAL during the comparison.

## Implementation details and inference speed

MAMMAL was developed with C++17 using Visual Studio Community 2017 on Windows 10 operating system. CUDA was utilized for accelerating silhouette energy term construction. Accuracy evaluation and behavior analysis were implemented with Python 3.7.9. Figures were plotted using the Matplotlib package of Python. On the BamaPig3D dataset, MAMMAL Detection stage took 50 ms per frame on single NVIDIA RTX 2080Ti GPU (11GB), MAMMAL Detection Matching stage took 0.15 ms per frame on CPU only, MAMMAL Mesh Fitting stage took 1200 ms per frame with GPU acceleration and 2000 ms without GPU support. In practice, 60 iterations were enough for $T = 0$, while 5 iterations per frame yielded fairly good results for $T > 0$.

## Statistics and Reproducibility

Suitable statistical method was used to predetermine the sample size. No data were excluded from the analyses. The experiments were randomized. The Investigators were blinded to allocation during experiments and outcome assessment. All values reported as mean ± SD unless otherwise stated. In box plots, boxes show median with IQR, with whiskers extending to 1.5 times the IQR, and with the arithmetic mean shown as a black square. Two group comparisons were analyzed using two-sided independent samples $t$-test (ttest_ind function in scipy). Data was considered significant at $p < 0.05$ (*), with $p < 0.01$ (**), $p < 0.001$ (***). Plot and analyses were performed in Visual Studio Community 2017 (with the aid of Std. C++17, Eigen3, OpenGL 4.5, OpenCV 4.5.0, Ceres 1.14) or Python 3.7.9 (with the aid of numpy 1.17.5, scipy 1.6.0, sklearn 0.24.1, skimage 0.18.1 and matplotlib 3.1.3). The mouse version of MAMMAL was developed in Python 3.9.12 and powered by PyTorch 1.12.1.

## Reporting summary

Further information on research design is available in the Nature Portfolio Reporting Summary linked to this article.

## Data availability

The data for PIG model is provided at https://github.com/anl13/PIG_model. The BamaPig2D and BamaPig3D datasets are available at https://github.com/anl13/MAMMAL_datasets. Data for evaluation and behavior analysis are released with the code. Data for figure generation are provided in the Source Data file. All the code can also be accessed at https://doi.org/10.17605/OSF.IO/F6JC5 (ref. [70]). Source data are provided with this paper.

## Code availability

The key code of MAMMAL written in C++ is released at https://github.com/anl13/MAMMAL_core. Codes for MAMMAL detection are released at https://github.com/anl13/pig_silhouette_det and https://github.com/anl13/pig_pose_det. Code and data related to evaluation and behavior analysis can be linked through https://github.com/anl13/MAMMAL_core. Code for mouse version of MAMMAL can be found at https://github.com/anl13/MAMMAL_mouse. All the code can also be accessed at https://doi.org/10.17605/OSF.IO/F6JC5 (ref. [70]). Software for generating Figs. 1a, b, e, 2a, e, f, g, 3j, 4g, 5b–h, i, Supplementary Fig. 2c, 5b, d is our self-developed renderer https://github.com/anl13/pig_renderer.

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

## Acknowledgements

This work is supported by the National Science Foundation of China project Nos. 62125107 (Y.B.L.), 61827805 (Y.B.L.), 92049114 (Y.C.J.), 31571097 (Y.C.J.), 81371361 (Y.C.J.), and 62171255 (T.Y.). It is also supported by Tsinghua-Peking Joint Center for Life Sciences, the Thousand-Talent Young Investigator Program, the IDG/McGovern Institute for Brain Research. It is also funded by the Guoqiang Institute of Tsinghua University (No. 2021GQG0001, T.Y.) and the Strategic Priority Research Program of the Chinese Academy of Sciences (XDA16030801toHT). We thank Beijing Sinogenetic Biotechnology Co., Ltd. for the help in the Beagle dog video recording.

## Author contributions

Y.B.L., Y.C.J., T.H., and L.A. conceived this project. Y.B.L., Y.C.J., and T.H. supervised this research. T.H. provided animals. L.A. and J.L.R. designed the hardware setup. J.L.R. and T.H. collected all the data. L.A. processed the data. L.A. developed the MAMMAL software and conducted experiments. T.Y. and Y.B.L. directed the development of the algorithms. T.Y., Y.B.L., T.H., and Y.C.J. gave critical discussions on the results. All authors participated in the writing of the paper.

## Competing interests

The authors declare no competing interests.
