## [Peer Review File · Nature Communications]

Three-dimensional surface motion capture of multiple freely moving pigs using MAMMALReviewer #1 (Remarks to the Author):

Key results

This paper presents a parameterized mesh model, MAMMAL, for tracking the poses and shapes of multiple pigs in a multi-camera video sequence. This model is used to persistently track different individual through a video sequence and distinguish between a range of behaviors, some defined in context of a single pig and the scene, e.g. drinking, some defined between a pair of pigs, e.g. head-tail interaction.

Validity and significance

The presented MAMMAL model and the process of fitting it to the video is well designed and well implemented and the results are excellent. The task of 3D-reconstructing several pigs is much more difficult than that of reconstructing a single animal.

However, the validity of the behavior recognition is low. The detected behaviors are very coarse and recognizable from the global relative and absolute positions and orientations of the pigs. No tail or ear motion behaviors - extremely important modes of communication for pigs - are included, and there is no record of whether the used behaviors have been selected by an expert in pig behaviors (an ethologist or veterinarian).

Data and methodology

The data collection and the 3D reconstruction of multiple pigs is of very high quality. As stated above however, the process of labeling behaviors has flaws. It should be noted that given the selected range of behaviors, the process of recognizing them from the reconstructed pig motion is well designed and implemented.

Analytical approach

This work is very much approached from a computer vision point of view. As described above, the aspects of the paper relating to the 3D reconstruction and action classification are of very high quality, while the ethological aspects of the problem formulation are lacking in several respects. In the introduction, the work is motivated with that the pig is an important model animal for research about human disease (which is indeed correct). The paper then skips a step in the motivation by saying that we then need to recognize pig behavior, without detailing what behavior needs to be recognized and why.

Suggested improvements

This reviewer would like to recommend a significant expansion of the first paragraph, where the animal welfare perspective is discussed - that animal behavior gives the caretaker cues about the health and wellbeing of the animals. And this is the responsibility of the caretaker.

Moreover, animal behavior can give us information about the cognition of the animal, which might be of interest in medical studies of dementia for example.

Third, the introduction should give account of the recent research of animal, in particular pig, behavior.

Fourth, I recommend involving a veterinarian or ethologist in the labeling.

Clarity and context

The paper is well written and well organized, except for some grammatical mistakes such as missing articles etc. Please let a native English speaker go through the text.

References

The authors are recommended to include references about the SMAL model by Silvia Zuffi, and also take a look at its species-specific developments such as BARC and hSMAL.

Reviewer expertise

The reviewer is an expert in visual human and animal tracking and behavior recognition.

Reviewer #2 (Remarks to the Author):

This manuscript describes a method to fit a pre-defined 3D pig shape to multi-view visual inputs, to study the social behavior of pigs. The 3D pig shape consists of 11239 vertices on the surface and 62 embedded joints. For training the machine learning models, this method requires multi-view inputs, a pre-defined 3D shape of a pig, as well as "bounding boxes, silhouettes, and keypoints of each individual" (line 95). The method consists of stages performing detection (pose estimation), identity matching, and mesh fitting, using model architectures from computer vision such as PointRend and HRNet for the first stage. To evaluate the method, the authors collect the BamaPig2D and BamaPig3D datasets with a set of manual keypoint annotations. Overall, this method is promising for studying pig behavior and I appreciate the authors for building a dataset of 2D and 3D pig poses; however, I would suggest the following be addressed before this manuscript can be considered for publication.

- My main concern about the paper is the ability of the approach to generalize to other experimental settings. All data is captured and evaluated by the authors in a single species in a single setting - while I do think the method is helpful for the specific analysis performed by the authors, I'm not sure if researchers in other labs will be able to reproduce or use this method. For example, the method requires building a detailed 3D mesh of the organism, with human annotated bounding boxes, silhouettes, and keypoints (compared to existing approaches mentioned by the authors on page 1 such as DLC and SLEAP, which requires significantly less human annotations to use, only 2D keypoints). Discussions or experiments by the authors on whether the tool is species-specific, scene-specific, or more general, would be helpful for assessing this work.

- I'd also appreciate some clarification from the authors on the evaluation procedure. For the evaluation procedure, the authors note that "the BamaPig3D and BamaPig2D datasets had no time point overlap" (line 184). How was the train and test split in each dataset created? My understanding is that both datasets consist of the same 4 pigs - would the model generalize to different sets of pigs? (Or different numbers of pigs?) How were model hyperparameters tuned (and how sensitive is the model to different hyperparameters)?

- It's not clear if the authors chose appropriate baselines for comparison to this approach. SLEAP (what seems to be the main baseline) is designed for 2D pose estimation, while this approach is designed for 3D + uses a pre-defined shape for pig, which contains a lot more human-defined information. The authors perform triangulation to obtain 3D keypoints from 2D. I'm wondering if a 3D tool like DANNCE (<https://www.nature.com/articles/s41592-021-01106-6>) or a computer vision method like BEV (https://openaccess.thecvf.com/content/CVPR2022/papers/Sun_Putting_People_in_Their_Place_Monocular_Regression_of_3D_People_CVPR_2022_paper.pdf) might be more appropriate as a baseline since they also leverage 3D information during training?

- The paper claims that "MAMMAL provides new possibilities for pig behavioral analysis" (line 113), with examples such as animal-scene interaction, posture discovery, or social behavior classification. However, it is not clear to me if these tasks cannot also be done with 3D pose estimation methods (ex: using SLEAP + triangulation, or DANNCE). 3D poses can also be used to measure distances to scene objects, map to features/representations for discovering postures, and compute distances between poses of agents. Discussions or experiments from the authors on the need for meshes would be helpful.

- Furthermore, I appreciate the authors for mentioning open-sourcing the code & data, but the Github links appear to be broken (I get a 404 error), so I cannot access the code & data.

Reviewer #3 (Remarks to the Author):

An et al. describe an approach to markerless pose estimation of multiple, interacting pigs by reconciling multiple camera views via a 3D mesh model. They built the PIG mesh model with articulated invisible keypoints, which is fit to multi-view video data after bounding-box detection and matching of animals. The paper characterizes the performance of this MAMMAL pipeline, evaluating it against ground-truth and comparing its performance with SLEAP in 2D.

Overall, I find the mesh-fitting idea to be an interesting one, and the paper does a good job executing it. Indeed, in the past, mesh-fitting in 3D has been a successful approach in some other areas of behavior quantification with computer vision where highly detailed reconstructions are needed, such as fine features of human facial expressions. My major concerns with the paper are with the depths of the analyses presented and the ease-of-use of the toolkit, as I elaborate below.

* The paper describes a motion capture pipeline for tracking multiple pigs and MAMMAL is a toolbox. However, while the C++ code for the analyses are supposed to be available on github, the links provided in the submitted manuscript are not live (I clicked on the links, but those repos are not found publicly). It is necessary to provide documentation, starter code, tutorials, or guides for new users who might be keen to try MAMMAL on their own data. If the paper is a toolbox paper, then I would expect the authors to supply the community with significantly more resources for potential users, which are vital to ensure the toolbox has wide impact. If the paper is not a toolbox paper, but rather more of a proof-of-principle paper, then I would expect the authors to demonstrate their algorithm works on more than one dataset, and ideally multiple species, to show its generalizability. Further, the quantitative behavior analysis pig behavior is relatively cursory.

* How would a potential end user of MAMMAL get started using this method tracking their own multiple animal data? Is the PIG model available for tracking different pigs, and how difficult would it be for another lab to build a mesh model for another animal, e.g. rats, mice, birds?

* How closely does the mesh model have to match the size and girth of the tracked animal? Specifically, is there any accounting in the algorithm for stretching the model (very small juvenile pigs, or very large pigs), or deforming portions of the mesh model (pig with an unusually large head, pregnant pig with large belly)? I would also like to see a quantification of the accuracy of the algorithm for animals of different sizes/ages.

* In Fig 3b, does 'invisible' keypoints mean that keypoint is not visible in any of the camera views? Or just some of them?

* The paper describes that 62 joints articular the full mesh model, but they only regressed 24 keypoints for motion reconstruction. I'm a bit confused by the fact that Fig 1b shows 19 keypoints, and Supp Video 1 shows the results from 62 keypoints also. How many keypoints are actually used in the results show?

Minor comments:

* There are many awkward sentences and phrases in the manuscript. The ideas in the paper would be more easily understood by the readers if the writing were more clear.

* The acronym for the mesh fitting system the author propose is MAMMAL, yet the paper only demonstrates its application on one species of mammals, namely pigs.

Reviewer #1 (Remarks to the Author):

Key results

This paper presents a parameterized mesh model, MAMMAL, for tracking the poses and shapes of multiple pigs in a multi-camera video sequence. This model is used to persistently track different individual through a video sequence and distinguish between a range of behaviors, some defined in context of a single pig and the scene, e.g. drinking, some defined between a pair of pigs, e.g. head-tail interaction.

Validity and significance

The presented MAMMAL model and the process of fitting it to the video is well designed and well implemented and the results are excellent. The task of 3D-reconstructing several pigs is much more difficult than that of reconstructing a single animal. However, the validity of the behavior recognition is low. The detected behaviors are very coarse and recognizable from the global relative and absolute positions and orientations of the pigs. No tail or ear motion behaviors - extremely important modes of communication for pigs - are included, and there is no record of whether the used behaviors have been selected by an expert in pig behaviors (an ethologist or veterinarian).

Thanks for this reviewer's comments. For the validity of the behavior recognition mentioned by this reviewer, four types of behaviors, including locomotion, postures, animal scene interaction, and animal social behaviors, were analyzed and demonstrated by MAMMAL in this study. These behaviors have been mainly concerned by previous 2D video-based pig behavior recognition research¹⁻¹⁷, while we are the first group to perform pig behavior analysis in a markerless 3D manner. As a proof-of-principle study at this stage, we did not focus on complex behavior analysis but pose classification¹⁸⁻²¹ and rule-based behavior analysis²² that have been well-established in mice. As this reviewer advised, we included the behavior recognition of pig tails (Fig. 4e and f), which have been proven to be closely related to different pig emotions^{1,6} and were also suggested by local ethologists from the Zoology Institute of Chinese Academy of Science.

We agree with this reviewer that tail/ear motion behaviors are extremely important modes for pig emotion and social communication^{1,6,7,13}. In the current submission, we employed MAMMAL to reconstruct the tail motions and quantified the motions in pigs with different social hierarchies^{2,6,7}. Specifically, we recorded videos of dominant pigs (n=4, with bigger body size) and subordinate pigs (n=4, with smaller body size) during their feeding (Fig. 4a). Our quantitative analysis revealed that the dominant pigs monopolized the trough (Fig. 4b and 4c). To quantify tail movements, we added two keypoints on pig tail (TailMid and TailTip, Fig. 4e and 4f) for 2D pose estimation and fit the tail motion with our PIG model. According to previous ethological study¹, passive hanging is more frequently observed in pigs that are exposed to an aversion situation, while loosely wagging is more related to positive emotions. In agreement with the previous report, we observed more loosely wagging but less passive hanging shown in the dominant pigs than that in the subordinate

pigs (Fig. 4g-j). The frequency of tail angle oscillations varies over the time, which was similar to the limb movements of mouse grooming; therefore, we adopted a previously-reported parameter PSD (power spectral density, PMID: 33875887) that was employed to reflect limb movements of mouse grooming to measure the pig tail angle oscillations (Fig. 4g and h). Within a 10-second time window (Fig. 4h), the PSD value (10.24) of the loosely wagging behavior peaked at 1.625Hz, which was much higher than that of the passive hanging behavior (PSD = 0.45, peaked at 0.125Hz), indicating the effectiveness of using PSD to classify tail behaviors (Fig. 4i). By automatically determining loosely wagging behavior whose PSD is higher than 1.5 across all the time windows, we found that the dominant pigs' tails oscillated significantly more than that of the subordinate pigs, which usually kept stationary (Fig. 4j). Therefore, the detailed behavior recognition by MAMMAL is proven to be valid at least in the tail movements in pigs with different social hierarchies.

Data and methodology

The data collection and the 3D reconstruction of multiple pigs is of very high quality. As stated above however, the process of labeling behaviors has flaws. It should be noted that given the selected range of behaviors, the process of recognizing them from the reconstructed pig motion is well designed and implemented.

Thanks for this reviewer's comments. To verify the behavior labeling, we consulted the local ethologists from the Zoology Institute of Chinese Academy of Science, and also referred the labelings to previously-annotated pig behaviors^{1-8,10,12-17}. As discussed above, we showed the general ability of MAMMAL to deal with locomotion, postures, animal-scene interactions, and social behaviors, which have covered most of the concerned behaviors by others¹⁻¹⁷. Some more specific behaviors can be tested in our system in the future.

Analytical approach

This work is very much approached from a computer vision point of view. As described above, the aspects of the paper relating to the 3D reconstruction and action classification are of very high quality, while the ethological aspects of the problem formulation are lacking in several respects.

In the introduction, the work is motivated with that the pig is an important model animal for research about human disease (which is indeed correct). The paper then skips a step in the motivation by saying that we then need to recognize pig behavior, without detailing what behavior needs to be recognized and why.

Thanks for this reviewer's comments. In the current submission, we elaborate more sentences on our motivations for pig behavior recognition in the Introduction (referred to 49-55 lines in this submission).

Suggested improvements

This reviewer would like to recommend a significant expansion of the first paragraph, where the animal welfare perspective is discussed - that animal behavior gives the caretaker cues about the health and wellbeing of the animals. And this is the responsibility of the caretaker.

Moreover, animal behavior can give us information about the cognition of the animal, which might be of interest in medical studies of dementia for example.

Third, the introduction should give account of the recent research of animal, in particular pig, behavior.

Fourth, I recommend involving a veterinarian or ethologist in the labeling.

Thanks for this reviewer's comments. We have followed this reviewer's instructions and revised the Introduction (referred to 44-55 lines in this submission).

Clarity and context

The paper is well written and well organized, except for some grammatical mistakes such as missing articles etc. Please let a native English speaker go through the text.

Thanks for this reviewer's comments. We have gone through the manuscript with a native English speaker to improve the language.

References

The authors are recommended to include references about the SMAL model by Silvia Zuffi, and also take a look at its species-specific developments such as BARC and hSMAL.

Thanks for this reviewer's comments. In the current submission, we include the references about the SMAL model²³⁻²⁵ and its species-specific developments such as BARC²⁶ and BITE²⁷ for dogs, and hSMAL²⁸ for horses.

Reviewer expertise

The reviewer is an expert in visual human and animal tracking and behavior recognition.

Reviewer #2 (Remarks to the Author):

This manuscript describes a method to fit a pre-defined 3D pig shape to multi-view visual inputs, to study the social behavior of pigs. The 3D pig shape consists of 11239 vertices on the surface and 62 embedded joints. For training the machine learning models, this method requires multi-view inputs, a pre-defined 3D shape of a pig, as well as "bounding boxes, silhouettes, and keypoints of each individual" (line 95). The method consists of stages performing detection (pose estimation), identity matching, and mesh fitting, using model architectures from computer vision such as PointRend and HRNet for the first stage. To evaluate the method, the authors collect the BamaPig2D and BamaPig3D datasets with a set of manual keypoint annotations. Overall, this method is promising for studying pig behavior and I appreciate the authors for building a dataset of 2D and 3D pig poses; however, I would suggest the following be addressed before this manuscript can be considered for publication.

- My main concern about the paper is the ability of the approach to generalize to other experimental settings. All data is captured and evaluated by the authors in a single species in a single setting - while I do think the method is helpful for the specific analysis performed by the authors, I'm not sure if researchers in other labs will be able to reproduce or use this method. For example, the method requires building a detailed 3D mesh of the organism, with human annotated bounding boxes, silhouettes, and keypoints (compared to existing approaches mentioned by the authors on page 1 such as DLC and SLEAP, which requires significantly less human annotations to use, only 2D keypoints). Discussions or experiments by the authors on whether the tool is species-specific, scene-specific, or more general, would be helpful for assessing this work.

Thanks for this reviewer's comments. In this revised manuscript, we demonstrate that MAMMAL can be applied to other species and experimental settings (Fig. 5, Supplementary Video 9, 10). For others to easily use MAMMAL as a toolbox, we made all our codes open-sourced, included instructions in Readme files of every code, summarized how to apply MAMMAL to multiple animals and other animal species in Supplementary Figure 10, and ensured that computational processing of MAMMAL can be handled by an undergraduate student with C++/Python coding experiences. Indeed, compared to 2D motion modeling, like DLC or SLEAP, MAMMAL needs more annotations, which we believe are affordable and worthwhile to generate 3D social motion with high quality. For example, for dog 3D motion recognition, it takes us only 7 hours for keypoint labeling, 10 hours for silhouette labeling, and 1.5 days for training process. In the future, we are going to make MAMMAL more friendly to users and compatible with more experimental settings.

- I'd also appreciate some clarification from the authors on the evaluation procedure. For the evaluation procedure, the authors note that "the BamaPig3D and BamaPig2D datasets had no time point overlap" (line 184). How was the train and test split in each dataset

created? My understanding is that both datasets consist of the same 4 pigs - would the model generalize to different sets of pigs? (Or different numbers of pigs?) How were model hyperparameters tuned (and how sensitive is the model to different hyperparameters)?

Thanks for this reviewer's comments. Our system did not use BamaPig3D for training but for testing. However, MAMMAL Detection was indeed trained on BamaPig2D for 2D silhouette and 2D pose estimation. Although BamaPig2D and BamaPig3D were derived from the same pigs, their images were extracted separately and did not share any overlaps. In the current submission, we provide evidence that MAMMAL can be applied for different settings that have never been annotated in BamaPig2D (Fig. 3d and 3e; Supplementary Video 6; Supplementary Video 8). Many hyperparameters, including the weights balancing different energy terms during mesh fitting, have been employed for finetuning the MAMMAL outputs. However, two hyperparameters we believe have a significant impact on MAMMAL outputs, including Tracking Distance and Silhouette Weight. Tracking Distance is the tracking threshold between 3D reconstruction results at time T-1 and 2D keypoints at time T, which has been used to constrain the allowed motion distance between two frames. Silhouette Weight reflects how the silhouettes can affect the mesh fitting results. If set Silhouette Weight to 0, only keypoints are used for mesh fitting, which allows flexible control of using silhouettes or not. The detailed hyperparameter settings were described in our current Methods and released codes.

- It's not clear if the authors chose appropriate baselines for comparison to this approach. SLEAP (what seems to be the main baseline) is designed for 2D pose estimation, while this approach is designed for 3D + uses a pre-defined shape for pig, which contains a lot more human-defined information. The authors perform triangulation to obtain 3D keypoints from 2D. I'm wondering if a 3D tool like DANNCE (<https://www.nature.com/articles/s41592-021-01106-6>) or a computer vision method like BEV (https://openaccess.thecvf.com/content/CVPR2022/papers/Sun_Putting_People_in_Their_Place_Monocular_Regression_of_3D_People_CVPR_2022_paper.pdf) might be more appropriate as a baseline since they also leverage 3D information during training?

Thanks for this reviewer's comments. In the current submission, we compared DANNCE with MAMMAL on mouse 3D motion recognition (Fig. 5a-e) and found that they performed similarly in estimating mouse extremities. We chose triangulation for comparison because it shares the same non-data-driven characteristics with MAMMAL and has been widely used in single animal 3D pose estimation²⁹⁻³². It is true that DANNCE^{33,34} or BEV³⁵ is good for data-driven baselines for 3D pose estimation. However, both DANNCE and BEV require large-scale 3D datasets for training. In fact, DANNCE was trained on a million scale dataset (Rat7M) and BEV was trained on Human3.6M³⁶, a million scale dataset collected by attaching markers on human bodies. However, it is difficult for us at this stage to collect such large-scale datasets for MAMMAL, which can be achieved in the future. We should clarify that we did not compare the whole MAMMAL system with SLEAP, but compared MAMMAL 2D pose

estimation with SLEAP. To avoid confusion, we moved the part to Supplementary Fig. 9b.

- The paper claims that "MAMMAL provides new possibilities for pig behavioral analysis" (line 113), with examples such as animal-scene interaction, posture discovery, or social behavior classification. However, it is not clear to me if these tasks cannot also be done with 3D pose estimation methods (ex: using SLEAP + triangulation, or DANNCE). 3D poses can also be used to measure distances to scene objects, map to features/representations for discovering postures, and compute distances between poses of agents. Discussions or experiments from the authors on the need for meshes would be helpful.

Thanks for this reviewer's comments. Although keypoints can be used for several behavioral analyses, they are sparse and lack the ability to measure dense surface contact between animals/agents. In contrast, articulated meshes are very valuable to encapsulate anatomical priors of animals^{37,38}, simplify the modeling of surface-to-surface contact (Fig. 4d), and serve as a medium to fuse multi-modal data like point clouds³⁹. As the articulated mesh models have achieved great success in human behavior modeling^{35,40-44} and clinical research⁴⁵, their applications in animal behavior capture are in the ascendant³⁹. We include this part in our current Discussion.

- Furthermore, I appreciate the authors for mentioning open-sourcing the code & data, but the Github links appear to be broken (I get a 404 error), so I cannot access the code & data.

Thanks for this reviewer's comments. We sincerely apologize for the unavailability of the links. In the current submission, we have fixed the problem.

Reviewer #3 (Remarks to the Author):

An et al. describe an approach to markerless pose estimation of multiple, interacting pigs by reconciling multiple camera views via a 3D mesh model. They built the PIG mesh model with articulated invisible keypoints, which is fit to multi-view video data after bounding-box detection and matching of animals. The paper characterizes the performance of this MAMMAL pipeline, evaluating it against ground-truth and comparing its performance with SLEAP in 2D.

Overall, I find the mesh-fitting idea to be an interesting one, and the paper does a good job executing it. Indeed, in the past, mesh-fitting in 3D has been a successful approach in some other areas of behavior quantification with computer vision where highly detailed reconstructions are needed, such as fine features of human facial expressions. My major concerns with the paper are with the depths of the analyses presented and the ease-of-use of the toolkit, as I elaborate below.

** The paper describes a motion capture pipeline for tracking multiple pigs and MAMMAL is a toolbox. However, while the C++ code for the analyses are supposed to be available on github, the links provided in the submitted manuscript are not live (I clicked on the links, but those repos are not found publicly). It is necessary to provide documentation, starter code, tutorials, or guides for new users who might be keen to try MAMMAL on their own data. If the paper is a toolbox paper, then I would expect the authors to supply the community with significantly more resources for potential users, which are vital to ensure the toolbox has wide impact. If the paper is not a toolbox paper, but rather more of a proof-of-principle paper, then I would expect the authors to demonstrate their algorithm works on more than one dataset, and ideally multiple species, to show its generalizability. Further, the quantitative behavior analysis pig behavior is relatively cursory.*

Thanks for this reviewer's comments. We sincerely apologize for the unavailability of the links. In this submission, we made all our codes open-sourced, included instructions in Readme files of every code, summarized how to apply MAMMAL to multiple animals and other animal species in Supplementary Figure 10, and ensured that computational processing of MAMMAL can be handled by an undergraduate student with C++/Python coding experiences. At the current stage, we believe, our paper is not only a proof-of-principle paper but also a toolbox with basic requirements. The different parts of MAMMAL work separately at present, therefore the users are required to have coding experiences in C++ and Python to implement all the codes without modifying them. More efforts are required to integrate them into a more user-friendly toolbox in the future. To generalize our algorithm, we further applied MAMMAL to mouse (Fig. 5a-e, Supplementary Video 9) and Beagle dogs (Fig. 5f-i, Supplementary Video 10). To improve quantitative behavior analysis, we analyzed the detailed tail motions of pigs with different social ranks in the current submission (Fig. 4).

** How would a potential end user of MAMMAL get started using this method tracking their own multiple animal data? Is the PIG model available for tracking different pigs, and how difficult would it be for another lab to build a mesh model for another animal, e.g. rats, mice, birds?*

Thanks for this reviewer's comments. In brief, one should first prepare synchronized multiple-view videos with camera calibration, then prepare 2D keypoints and silhouettes using 2D detectors (e.g. DLC, SLEAP, HRNet, PointRend, or any other method). Afterwards, one needs to prepare a mesh model for the target species, and manually define the mapping between 2D keypoints and articulated mesh joints or vertices. With all the above data available, one could run the MAMMAL algorithm following the instruction we described in Method. Although our PIG model is designed for Bama mini pig, it is feasible to apply the PIG model for other pig strains, like *Sus scrofa*. To further improve the tracking accuracy of other pig strains rather than Bama pig, one could tune the specifications (e.g. bone lengths) of our PIG model to accommodate to specific pig shape using animation softwares, like MAYA or Blender. To build mesh models for other customized animals, we recommend purchasing an off-the-shelf animation model (e.g. from <https://sketchfab.com/>) or selecting an open-sourced model (e.g. SMAL²⁵) and modifying it by using MAYA or Blender according to endusers their own requirements.

** How closely does the mesh model have to match the size and girth of the tracked animal? Specifically, is there any accounting in the algorithm for stretching the model (very small juvenile pigs, or very large pigs), or deforming portions of the mesh model (pig with an unusually large head, pregnant pig with large belly)? I would also like to see a quantification of the accuracy of the algorithm for animals of different sizes/ages.*

Thanks for this reviewer's comments. MAMMAL is robust to different shapes of pigs, and the algorithm automatically tunes scale parameter to control the model shape. Although we did not stretch the model, we added the quantification of the accuracy for tracking pigs with different sizes in this revised manuscript (Fig. 3d and 3e), which demonstrates that MAMMAL can achieve similar accuracy for very small or very large pigs. In the future, we expect to build a linear blend shape model for pig similar to SMPL⁴² for human or SMAL²⁵ for quadrupeds.

** In Fig 3b, does 'invisible' keypoints mean that keypoint is not visible in any of the camera views? Or just some of them?*

Thanks for this reviewer's comments. The invisible keypoints are keypoints visible to no more than one view (Fig. 3b). To assess the tracking accuracy of invisible ones, we labeled invisible keypoints with the help of the PIG model (Supplementary Fig. 8).

** The paper describes that 62 joints articular the full mesh model, but they only regressed 24 keypoints for motion reconstruction. I'm a bit confused by the fact that Fig 1b shows 19*

keypoints, and Supp Video 1 shows the results from 62 keypoints also. How many keypoints are actually used in the results show?

Thanks for this reviewer's comments. Here we would like to clarify some terms used in this manuscript. First, 62 joints, also named articulated joints, represent the ultimate motion freedom of the PIG model. Each joint is controlled by a 3 DOF (degrees of freedom) rotation vector, the changes of which would affect the geometric position of both joints and vertices of the PIG model. Second, 24 crucial joints (a subset of 62 joints) were used for trunk/leg motion control by ignoring motions on the tail, ears, or toes, which is more efficient for trunk social contact without loss of accuracy. For detailed tail motion capture in the current submission, we utilized all the 62 joints, including the tail, for motion reconstruction (Fig. 4, Supplementary Video 8). Third, 19 3D keypoints are mapped from the joints or vertices of the mesh model, and correspond to the ones labeled for 2D pose estimation detection. Note that a 3D keypoint can be the position of a joint or a vertex, or the interpolation of several joints/vertices.

Minor comments:

** There are many awkward sentences and phrases in the manuscript. The ideas in the paper would be more easily understood by the readers if the writing were more clear.*

Thanks for this reviewer's comments. In the current submission, we asked a native speaker to improve our writing.

** The acronym for the mesh fitting system the author propose is MAMMAL, yet the paper only demonstrates its application on one species of mammals, namely pigs.*

Thanks for this reviewer's comments. In this submission, we demonstrate that MAMMAL can be applied to other mammals, including mouse (Fig. 5a-e, Supplementary Video 9) and Beagle dogs (Fig. 5f-i, Supplementary Video 10).

References:

- 1 Camerlink, I. & Ursinus, W. W. Tail postures and tail motion in pigs: A review. *Appl Anim Behav Sci* **230**, doi:ARTN 10507910.1016/j.applanim.2020.105079 (2020).
- 2 Chen, D. *et al.* Multi-breed investigation of pig social rank and biological rhythm based on feeding behaviors at electronic feeding stations. *Livest Sci* **245**, doi:ARTN 10441910.1016/j.livsci.2021.104419 (2021).
- 3 D'Eath, R. B. *et al.* Changes in tail posture detected by a 3D machine vision system are associated with injury from damaging behaviours and ill health on commercial pig farms. *Plos One* **16**, doi:ARTN e025889510.1371/journal.pone.0258895 (2021).
- 4 Gan, H. M. *et al.* Fast and accurate detection of lactating sow nursing behavior with CNN-based optical flow and features. *Comput Electron Agr* **189**, doi:ARTN 10638410.1016/j.compag.2021.106384 (2021).
- 5 Gan, H. M. *et al.* Automated detection and analysis of social behaviors among preweaning piglets using key point-based spatial and temporal features. *Comput Electron Agr* **188**, doi:ARTN 10635710.1016/j.compag.2021.106357 (2021).
- 6 Houpt, K. A. *Domestic animal behavior for veterinarians and animal scientists*. 6th edition. edn, (John Wiley & Sons, Inc.,, 2018).
- 7 Jensen, P. & Woodgush, D. G. M. Social Interactions in a Group of Free-Ranging Sows. *Appl Anim Behav Sci* **12**, 327-337, doi:Doi 10.1016/0168-1591(84)90125-4 (1984).
- 8 Ji, Y. P., Yang, Y. & Liu, G. Recognition of Pig Eating and Drinking Behavior Based on Visible Spectrum and YOLOv2. *Spectrosc Spect Anal* **40**, 1588-1594, doi:10.3964/j.issn.1000-0593(2020)05-1588-07 (2020).
- 9 Kittawornrat, A. & Zimmerman, J. J. Toward a better understanding of pig behavior and pig welfare. *Anim Health Res Rev* **12**, 25-32, doi:10.1017/S1466252310000174 (2011).
- 10 Kuster, S. *et al.* Usage of computer vision analysis for automatic detection of activity changes in sows during final gestation. *Comput Electron Agr* **169**, doi:ARTN 10517710.1016/j.compag.2019.105177 (2020).
- 11 Matthews, S. G., Miller, A. L., Clapp, J., Plotz, T. & Kyriazakis, I. Early detection of health and welfare compromises through automated detection of behavioural changes in pigs. *Vet J* **217**, 43-51, doi:10.1016/j.tvjl.2016.09.005 (2016).
- 12 Mills, D. *Domestic Animal Behaviour for Veterinarians and Animal Scientists*, Sixth Edition. *Anim Welfare* **28**, 234-235 (2019).
- 13 Reimert, I., Bolhuis, J. E., Kemp, B. & Rodenburg, T. B. Indicators of positive and negative emotions and emotional contagion in pigs. *Physiol Behav* **109**, 42-50, doi:10.1016/j.physbeh.2012.11.002 (2013).
- 14 Wang, S. L. *et al.* The Research Progress of Vision-Based Artificial Intelligence in Smart Pig Farming. *Sensors-Basel* **22**, doi:ARTN 654110.3390/s22176541 (2022).
- 15 Yang, Q. M. & Xiao, D. Q. A review of video-based pig behavior recognition. *Appl Anim Behav Sci* **233**, doi:ARTN 10514610.1016/j.applanim.2020.105146 (2020).
- 16 Zhang, K. F., Li, D., Huang, J. Y. & Chen, Y. F. Automated Video Behavior Recognition of Pigs Using Two-Stream Convolutional Networks. *Sensors-Basel* **20**, doi:ARTN 108510.3390/s20041085 (2020).

- 17 Zheng, C. *et al.* Automatic recognition of lactating sow postures from depth images by
deep learning detector. *Comput Electron Agr* **147**, 51-63,
doi:10.1016/j.compag.2018.01.023 (2018).
- 18 Lauer, J. *et al.* Multi-animal pose estimation, identification and tracking with DeepLabCut.
Nat Methods **19**, doi:10.1038/s41592-022-01443-0 (2022).
- 19 Mathis, A. *et al.* DeepLabCut: markerless pose estimation of user-defined body parts with
deep learning. *Nat Neurosci* **21**, 1281-+, doi:10.1038/s41593-018-0209-y (2018).
- 20 Pereira, T. D. *et al.* Fast animal pose estimation using deep neural networks. *Nat Methods*
16, 117-+ (2019).
- 21 Pereira, T. D. *et al.* SLEAP: A deep learning system for multi-animal pose tracking. *Nat*
Methods **19**, 486-+ (2022).
- 22 de Chaumont, F. *et al.* Real-time analysis of the behaviour of groups of mice via a depth-
sensing camera and machine learning. *Nat Biomed Eng* **3**, 930-942, doi:10.1038/s41551-
019-0396-1 (2019).
- 23 Zuffi, S., Kanazawa, A., Berger-Wolf, T. & Black, M. J. Three-D Safari: Learning to Estimate
Zebra Pose, Shape, and Texture from Images" In the Wild". *Proceedings of the IEEE/CVF*
International Conference on Computer Vision, 5359-5368 (2019).
- 24 Zuffi, S., Kanazawa, A. & Black, M. J. in *Proceedings of the IEEE conference on Computer*
Vision and Pattern Recognition. 3955-3963.
- 25 Zuffi, S., Kanazawa, A., Jacobs, D. W. & Black, M. J. 3D menagerie: Modeling the 3D shape
and pose of animals. *Proceedings of the IEEE conference on computer vision and pattern*
recognition, 6365-6373 (2017).
- 26 Rueegg, N., Zuffi, S., Schindler, K. & Black, M. J. BARC: Breed-Augmented Regression
Using Classification for 3D Dog Reconstruction from Images. *International Journal of*
Computer Vision, doi:10.1007/s11263-023-01780-3 (2023).
- 27 Ruegg, N., Tripathi, S., Schindler, K., Black, M. J. & Zuffi, S. BITE: Beyond Priors for Improved
Three-D Dog Pose Estimation. *Proceedings of the IEEE conference on computer vision*
and pattern recognition (2023).
- 28 Li, C. *et al.* in *CV4Animals Workshop, Proceedings IEEE Conference on Computer Vision*
and Pattern Recognition (CVPR).
- 29 Karashchuk, P. *et al.* Anipose: A toolkit for robust markerless 3D pose estimation. *Cell Rep*
36, doi:ARTN 10973010.1016/j.celrep.2021.109730 (2021).
- 30 Huang, K. *et al.* A hierarchical 3D-motion learning framework for animal spontaneous
behavior mapping. *Nat Commun* **12**, doi:ARTN 278410.1038/s41467-021-22970-y (2021).
- 31 Gunel, S. *et al.* DeepFly3D, a deep learning-based approach for 3D limb and appendage
tracking in tethered, adult *Drosophila*. *Elife* **8**, doi:ARTN e4857110.7554/eLife.48571
(2019).
- 32 Nath, T. *et al.* Using DeepLabCut for 3D markerless pose estimation across species and
behaviors. *Nat Protoc* **14**, 2152-2176, doi:10.1038/s41596-019-0176-0 (2019).
- 33 Li, T., Severson, K. S., Wang, F. & Dunn, T. W. Improved 3D Markerless Mouse Pose
Estimation Using Temporal Semi-supervision. *International Journal of Computer Vision*,
1-17 (2023).
- 34 Dunn, T. W. *et al.* Geometric deep learning enables 3D kinematic profiling across species
and environments. *Nat Methods* **18**, 564-+, doi:10.1038/s41592-021-01106-6 (2021).

- 35 Sun, Y. *et al.* Putting People in their Place: Monocular Regression of 3D People in Depth. *Proc Cvpr Ieee*, 13233-13242, doi:10.1109/Cvpr52688.2022.01289 (2022).
- 36 Ionescu, C., Papava, D., Olaru, V. & Sminchisescu, C. Human3.6M: Large Scale Datasets and Predictive Methods for 3D Human Sensing in Natural Environments. *IEEE Trans Pattern Anal Mach Intell* **36**, 1325-1339, doi:10.1109/TPAMI.2013.248 (2014).
- 37 Bolanos, L. A. *et al.* A three-dimensional virtual mouse generates synthetic training data for behavioral analysis. *Nat Methods* **18**, 378-381, doi:10.1038/s41592-021-01103-9 (2021).
- 38 Lobato-Rios, V. *et al.* NeuroMechFly, a neuromechanical model of adult *Drosophila melanogaster*. *Nat Methods* **19**, 620-+, doi:10.1038/s41592-022-01466-7 (2022).
- 39 Bohoslav, J. P. *et al.* ArMo: An Articulated Mesh Approach for Mouse 3D Reconstruction. *bioRxiv*, doi:10.1101/2023.02.17.526719 (2023).
- 40 Blanz, V. & Vetter, T. A morphable model for the synthesis of 3D faces. *Comp Graph*, 187-194, doi:Doi 10.1145/311535.311556 (1999).
- 41 Joo, H., Simon, T. & Sheikh, Y. Total Capture: A 3D Deformation Model for Tracking Faces, Hands, and Bodies. *2018 Ieee/Cvf Conference on Computer Vision and Pattern Recognition (Cvpr)*, 8320-8329, doi:10.1109/Cvpr.2018.00868 (2018).
- 42 Loper, M., Mahmood, N., Romero, J., Pons-Moll, G. & Black, M. J. SMPL: A Skinned Multi-Person Linear Model. *Acm T Graphic* **34**, doi:Artn 24810.1145/2816795.2818013 (2015).
- 43 Zhang, Y. X. *et al.* 4D Association Graph for Realtime Multi-person Motion Capture Using Multiple Video Cameras. *2020 Ieee/Cvf Conference on Computer Vision and Pattern Recognition (Cvpr)*, 1321-1330, doi:10.1109/Cvpr42600.2020.00140 (2020).
- 44 Zhang, Y. X. *et al.* Lightweight Multi-person Total Motion Capture Using Sparse Multi-view Cameras. *2021 Ieee/Cvf International Conference on Computer Vision (Iccv 2021)*, 5540-5549, doi:10.1109/Iccv48922.2021.00551 (2021).
- 45 Wang, R. X. & Lin, H. T. Anonymizing facial images to improve patient privacy. *Nat Med* **28**, 1767-1768, doi:10.1038/s41591-022-01967-0 (2022).

Reviewer #1 (Remarks to the Author):

Thank you for your thorough revision of the manuscript. The authors address my concerns about ethological validity well, e.g. by introducing the new section "Using MAMMAL to quantify the tail motion for pigs in different social hierarchies" in lines 201-223, as well as Figure 4.

My other concerns about labeling, motivation, language are also addressed in a satisfactory manner.

I therefore recommend the paper for acceptance.

Reviewer #2 (Remarks to the Author):

I would like to thank the authors for responding to the reviewer's concerns. The updated manuscript looks much better to me and have addressed many of my comments. My main remaining comments are the time it requires to use this tool and quantitative evaluations. The authors mention that "Indeed, compared to 2D motion modeling, like DLC or SLEAP, MAMMAL needs more annotations, which we believe are affordable and worthwhile to generate 3D social motion with high quality. For example, for dog 3D motion recognition, it takes us only 7 hours for keypoint labeling, 10 hours for silhouette labeling, and 1.5 days for training process." But my understanding is that the model still needs an input 3D mesh (in addition to those other requirements)? How long did it take to construct the mesh in Figure 5b or 5g? What about organisms where we don't have the 3D mesh or if the mesh does not fit the shape of the organism well?

The authors included qualitative results in the figures and videos, but limited quantitative results with comparisons against other baselines (ex: Figure 5e shows a comparison against DANNCE, where the methods perform similarly, except DANNCE is worse on the tail). It would be great if the authors could discuss this in more detail, given that quantitative numbers against baselines are similar.

Ultimately, the paper would be stronger with more quantitative comparisons, but as it's current form, it already shows more promising results compared to the first submission and the open-source code may already be useful to some labs with 3D meshes.

REVIEWER COMMENTS

Reviewer #2 (Remarks to the Author):

I would like to thank the authors for responding to the reviewer's concerns. The updated manuscript looks much better to me and have addressed many of my comments. My main remaining comments are the time it requires to use this tool and quantitative evaluations. The authors mention that "Indeed, compared to 2D motion modeling, like DLC or SLEAP, MAMMAL needs more annotations, which we believe are affordable and worthwhile to generate 3D social motion with high quality. For example, for dog 3D motion recognition, it takes us only 7 hours for keypoint labeling, 10 hours for silhouette labeling, and 1.5 days for training process." But my understanding is that the model still needs an input 3D mesh (in addition to those other requirements)? How long did it take to construct the mesh in Figure 5b or 5g? What about organisms where we don't have the 3D mesh or if the mesh does not fit the shape of the organism well?

Thanks for the reviewer's comments. As this Reviewer mentioned, the model still needs an input 3D mesh. For Figure 5b, we spent one day to extract the mouse mesh model from original blender file provided in a previous publication¹. More specifically, we exported the vertices, skinning weights, and embedded joints from the original Blender file; we manually removed the whiskers using Blender and Meshlab; and we defined keypoints on it (L.230 in main text, L.525-529 in Methods). For Fig 5h, we modified a commercial beagle dog mesh model by using MAYA software, which took us about one day. More specifically, we removed whiskers and useless bones and defined keypoints on it (L.255 in main text, L.554-556 in Methods). For an animal species that does not have the 3D mesh, we recommend to follow the process as previously described¹ to create a customized 3D mesh model. If the mesh model does not fit the shape of the animal species well, we recommend to tune the mesh model according to the captured images using MAYA software or automatically deform the mesh vertices using non-rigid deform algorithms according to the captured silhouettes (L.340-345 in Methods). Even when the mesh model does not strictly fit the target animal, our method is robust enough to provide useful results.

The authors included qualitative results in the figures and videos, but limited quantitative results with comparisons against other baselines (ex: Figure 5e shows a comparison against DANNCE, where the methods perform similarly, except DANNCE is worse on the tail). It would be great if the authors could discuss this in more detail, given that quantitative numbers against baselines are similar.

Ultimately, the paper would be stronger with more quantitative comparisons, but as it's current form, it already shows more promising results compared to the first submission and the open-source code may already be useful to some labs with 3D meshes.

Thanks for the reviewer's comments. In current submission, we performed more quantitative comparisons with DANNCE-T on different view settings (Fig. 5e and 5f, and corresponding

L.237-245 in main text and L.546-550 in Methods), and added comparison to VoxelPose ², which is a strong baseline for multiple human 3D pose estimation (Fig. 5j and 5k, and corresponding L.256-265 in main text and L.561-568 in Methods). Although the quantitative numbers against DANNCE look similarly good by inferring on total views, we argue that such volume-based methods (both DANNCE and VoxelPose) sacrifice generalization in exchange for accuracy. More specifically, their performance would degrade drastically when inferring on different camera settings (e.g. less views) without re-training on the target camera setting. In Fig. 5e, 5f, 5j, and 5k, we performed view study on DANNCE-T for mouse and VoxelPose for Beagle dogs, and proved that MAMMAL was more robust to different camera settings compared to DANNCE-T and VoxelPose. Moreover, except “tail” point on which DANNCE-T had bad performance, the average error of MAMMAL (2.20 ± 1.25 mm) was still lower than that of DANNCE-T (2.71 ± 3.59 mm).

References:

- 1 Bolanos, L. A. *et al.* A three-dimensional virtual mouse generates synthetic training data for behavioral analysis. *Nat Methods* **18**, 378–381 (2021). <https://doi.org/10.1038/s41592-021-01103-9>
- 2 Tu, H., Wang, C. & Zeng, W. in *Computer Vision—ECCV 2020: 16th European Conference, Glasgow, UK, August 23–28, 2020, Proceedings, Part I 16*. 197–212 (Springer).